# Common and distinct neurofunctional signatures of dynamic naturalistic emotion regulation strategies

Heng Jiang [1,2], Jingxian He[1,2], Kaeli Zimmermann[3], Xinqi Zhou [4], Xianyang Gan[1,2], Stefania Ferraro[1,2], Lan Wang[1,2], Bo Zhou[1,2], Liyuan Li[1,2], Keith M. Kendrick [1,2], Weihua Zhao [1,2], Dezhong Yao [1,2], Tifei Yuan [5,6], Feng Zhou [7,8] ✉ & Benjamin Becker [9,10] ✉

Adaptive emotion regulation is essential for mental health. Reappraisal and acceptance are effective yet cognitively distinct emotion regulation strategies. A key unanswered question is whether their neural implementations are supported by common overarching or distinct neurofunctional processes, especially under dynamic, naturalistic conditions that mirror real-life scenarios. Here, we combined naturalistic fMRI with multivariate predictive modeling to develop neurofunctional signatures that accurately and comprehensively characterize negative affect and its regulation via acceptance and reappraisal in dynamic, immersive contexts ($n = 59$). These signatures demonstrated process-specificity and generalizability across cohorts, cultures, and modalities ($n = 33$, 358, 45, and 33, respectively). Emotion regulation strategies were encoded in distributed, distinguishable neural representations, with shared contributions from the default mode network and strategy-specific contributions from the amygdala, somatomotor and attention (acceptance), and the frontoparietal control (reappraisal) networks. The neuromarkers precisely identified strategy-specific ER impairments in male cannabis users ($n_{healthy\_controls} = 48$, $n_{cannabis\_users} = 49$), underscoring their potential clinical translational relevance. Collectively, these findings demonstrate shared and distinct neural signatures of reappraisal and acceptance, highlight the critical role of whole-brain integration in emotion regulation, and provide comprehensive, clinically relevant brain models of emotion regulation and dysregulation in naturalistic contexts.

Emotion regulation (ER)—the ability to modulate the experience and expression of emotions—is fundamental to adaptive functioning and mental health[1–6]. Impairments in ER represent a core transdiagnostic feature across major mental disorders (MDs)[5,7–11]. Regulation of negative affect can be achieved by several mental strategies, with reappraisal and acceptance being particularly effective[2,12–15]. Both are central building blocks of psychotherapeutic

interventions[16–18] and highly suitable for managing everyday emotional challenges[19,20].

While both strategies focus on cognitive changes[15] (see *process model of emotion regulation*[1,2]), they differ markedly in their operational and cognitive mechanisms. Reappraisal is based on reinterpreting the subjective meaning of the emotion-inducing event to control its affective impact[21,22]. It critically relies on multiple executive

functions, including attention, working memory, and cognitive control, which represent an active and cognitively demanding process[23,24]. While reappraisal is the most extensively studied ER strategy, a critical question remains: what constitutes an effective and psychologically distinct alternative? Strategies like avoidance, suppression, or distraction have been commonly investigated; however, they are often maladaptive and associated with poor long-term outcomes[6,7]. In contrast, acceptance, a core component of mindfulness-based third-wave cognitive therapies, involves nonjudgmental awareness of emotional experiences without attempts to control them[25,26]. It has been recognized as both a potent and adaptive alternative, and is considered a less cognitively demanding and more passive strategy relying on self-referential processes and metacognitive awareness[24]. This distinction is crucial for both cognitive neuroscience and therapeutic applications. If reappraisal and acceptance simply recruit the same neural pathways, it would suggest a final common pathway for effective regulation. Conversely, dissociable neural patterns would provide strong evidence for multiple, distinct routes to ER and adaptive mental health, with significant implications for personalized interventions. Additionally, ongoing debates[27–29] about the extent of neural overlap between emotion generation and regulation (as well as the overlap between different emotion regulation strategies[30]) further highlight the need for empirical work that initially carefully characterizes the neural basis of distinct emotion regulation processes. Directly comparing reappraisal and acceptance represents as such a critical step for characterizing strategy-specific neural substrates involved in the regulation of negative affect.

The prevailing neurobiological models of emotion distinguish between brain systems involved in emotional reactivity, e.g., the amygdala, insula, and systems involved in regulation, including lateral prefrontal and parietal regions[31–33]. fMRI meta-analyses indicate that both strategies effectively modulate activity in subcortical emotion "reactivity" regions, e.g., the amygdala[26,34], while enhancing the engagement of ventrolateral prefrontal regulatory regions (vlPFC)[30,35]. Notably, reappraisal is strongly associated with engagement of the fronto-parietal control network[32,33,36], reflecting its reliance on regulatory cognitive processes, while acceptance is associated with engagement of core regions of the posterior midline default mode network (DMN)[24,26,37–39], suggesting a distinct neural signature reflective of self-related and introspective processing. Together, further research is required to better characterize and extend theoretical models describing the relationship between emotion generation and regulation. This is particularly important given the distinct cognitive mechanisms and implementation demands that underlie acceptance and reappraisal.

Despite extensive work using conventional neuroimaging and meta-analytic approaches several key questions remain unresolved, including (1) a precise and comprehensive mapping of the common and distinct whole-brain neural representations of reappraisal and acceptance; (2) the generalizability of the identified neural systems that have been extensively assessed using sparsely presented static stimuli (e.g. pictures) to naturalistic, dynamic emotional experiences which more closely reflect real-world affective experiences; (3) the extent to which ER relies on localized modules versus distributed neural computations across the whole brain; and (4) the translational potential of these neural representations as precise neuromarkers for ER impairments in clinical population.

First, traditional mass-univariate analysis, widely used in previous studies, presents several limitations with respect to establishing comprehensive and accurate brain models for specific mental processes, including (1) voxel-wise independence assumptions that reduce sensitivity to comprehensively determine cognitive functions[40] and (2) the frequent use of ROIs-based averaging, in which activation signals are averaged across voxels (each containing a massive amount of neurons, ~5.5 million) for group-level inference, which may yield nonspecific signals[41,42].

Second, few studies[24,35,38] have directly compared acceptance and reappraisal using a within-subject design or under ecologically valid conditions. Most neurobiologically-informed models instead rely on meta-analyses[5,43,44], which are limited by the univariate approach of the original studies and further constrained by variability indicated by preprocessing protocols[45]. Moreover, these studies used sparsely presented, static, and isolated stimuli (e.g., affective pictures), which limits ecological validity[46,47], in particular in the context of emerging evidence indicating that neural systems that support affective experience differ substantially when engaged during dynamic naturalistic experiences[48–51].

Third, conventional analytical approaches, which often rely on certain significance thresholds to identify activated brain regions, may have led many previous studies and theoretical frameworks to emphasize the contributions of specific brain areas or functional networks. However, emerging evidence from both human and animal research indicates that complex psychological processes (e.g., decision-making[52], responses to disgust[53], fear[54], or pain[55]) involve distributed neural computations that integrate information from the entire brain. Whether acceptance and reappraisal are similarly encoded at a whole-brain level remains an open empirical question.

Fourth, despite the recognition of impaired ER as a transdiagnostic impairment in MDs, including addiction[5,7–11], findings have yet to be translated into clinically useful neuromarkers. While initial progress has been made in developing a neuromarker for cue reactivity, in the domain of ER is lacking[56].

To address these critical limitations, we developed and extensively evaluated three comprehensive and ecologically valid brain models under naturalistic conditions, capturing negative emotional experience, acceptance, and reappraisal, with substantial potential for clinical application. We here capitalize on the combination of functional magnetic resonance imaging (fMRI) with machine-learning-based multivariate pattern analysis (MVPA) and naturalistic paradigms[57–60]. MVPA leverages information across multiple spatial scales[41,61] to detect distributed patterns across the entire brain, which is uniquely suited to quantifying the overlap and dissociation among psychological processes without relying on a priori assumptions regarding "emotion generation" versus "regulation" regions. It further enables the development of comprehensive, accurate, and generalizable whole-brain models of emotional experiences ('neuroaffective signatures') with improved sensitivity and specificity. Importantly, MVPA offers the opportunity to compare predictive performances across networks or regions of interest while controlling for confounds related to voxel number. These features collectively enhance the translation potential of MVPA-derived models as clinically useful neuromarkers for MDs[56,61,62]. To further increase the ecological validity of these models, recent studies have employed naturalistic paradigms with dynamic videos, speech, and music as materials that are consistent with the typical sensory stimuli encountered in realistic daily life[46–48,50,51,63,64].

In this work, we aimed to precisely determine common and distinct neural representations of reappraisal and acceptance during naturalistic dynamic emotional processing combined with MVPA-based neural decoding. Briefly, 59 participants from a pre-registered discovery cohort and 33 participants from a pre-registered validation cohort underwent a dynamic naturalistic fMRI ER paradigm (Fig. 1a), with acceptance (NA), reappraisal (NR), or natural emotional experience to negative (NV) or neutral clips (NeutV). We systematically tested (1) whether it is possible to develop precise and robust neural signatures to determine negative experiences (naturalistic negative emotional signature, NNES) and their regulation via reappraisal (naturalistic emotion regulation signature-reappraisal, NERS-R) and acceptance (naturalistic emotion regulation signature-acceptance,

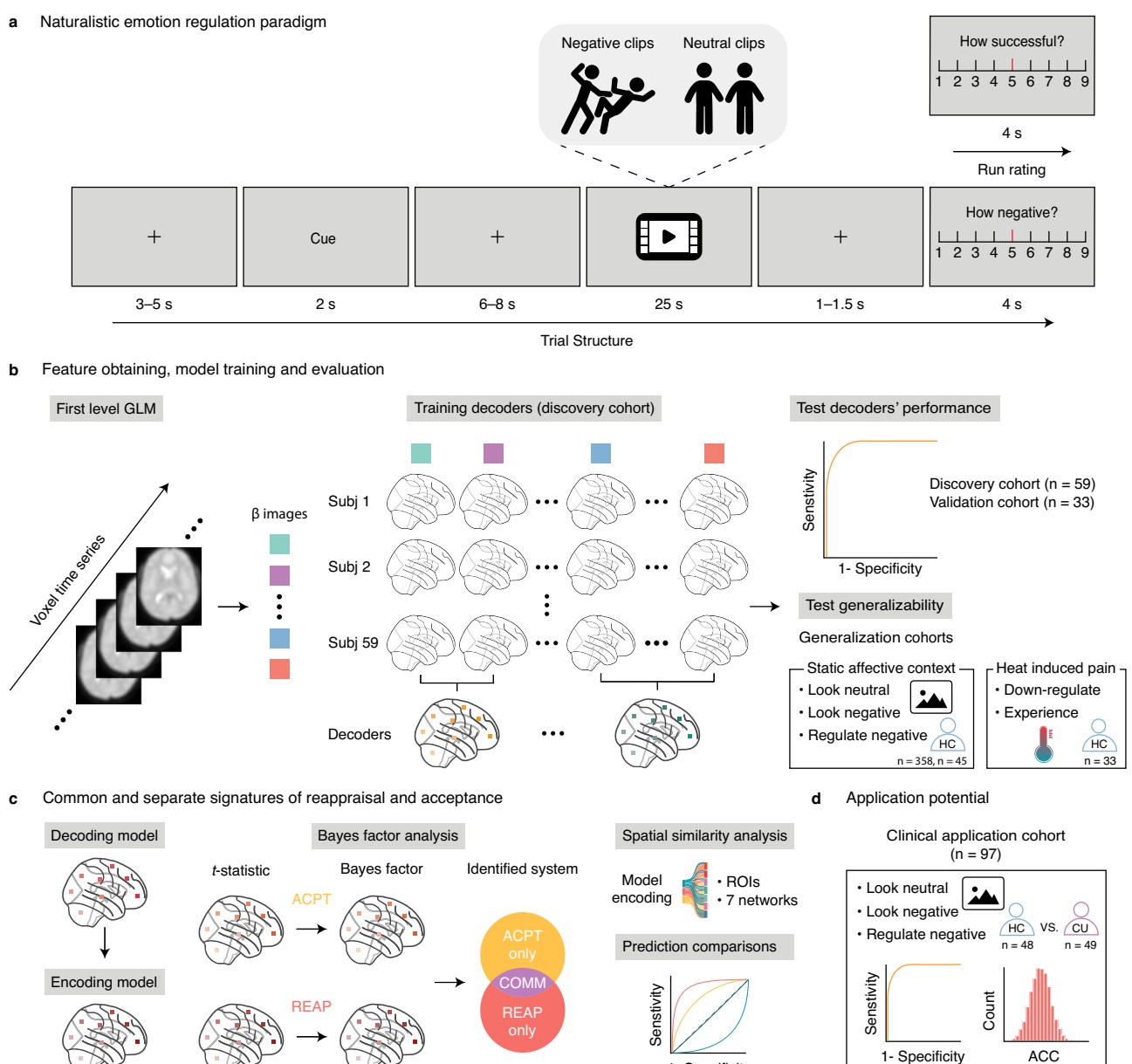

**Fig. 1 | Task paradigm, model evaluation, and general analytic workflow.**
**a** Naturalistic emotion regulation paradigm used in the discovery and validation cohort. The cues were "react", "acceptance", or "reappraisal", indicating which strategy participants should use to view the neutral/negative video clips. At the end of each trial, participants rated their negative feelings. At the end of each run, participants rated their average success level in this run. Notably, the schematics were used only for illustrative purposes to avoid copyright issues and were not part of the original stimulus set. **b** Feature estimation, model training, and evaluation. First-level contrast maps were used as features in the prediction analysis. The whole brain multivariate pattern predictive of the acceptance, reappraisal, or reaction to the naturalistic stimulus was trained on the discovery sample ($n = 59$) using the support vector machine (SVM) and further evaluated in discovery (cross-validated), validation ($n = 33$) cohorts. The generalizability of the developed signatures across samples, MRI systems, culture, and stimuli (dynamic versus static[34,65], $n = 358$ and n = 45, or heat-induced pain[66], $n = 33$). **c** Systematically investigate the common and distinct neuro-basis of acceptance and reappraisal. Determining the contribution of specific brain systems by using univariate and multivariate methods. The performance of isolated brain regions or systems was tested by multiple prediction analyses. **d** Testing the specificity of developed models in terms of distinguishing corresponding mental processing from other decoders, e.g., cannabis users (CU, $n = 49$) and healthy controls (HC, $n = 48$)[71]. Subj subject, ACPT acceptance, REAPP reappraisal, COMM common, ROIs regions-of-interest, ACC accuracy.

NERS-A), respectively, across the discovery cohort and independent validation cohort (Fig. 1b); (2) whether the developed decoders generalize to independent ER processing investigated using conventional static paradigms (e.g., picture) in two independent dataset of healthy individuals ($n = 358$, $n = 45$)[34,65] and reappraisal pain induced by heat stimuli ($n = 33$)[66]. Combining multiple analysis methods to determine (3) the distinct and common brain representations and systems that underlie acceptance and reappraisal, utilizing, e.g., Bayes Factor (BF) analysis, which allows examining for both the null and alternative hypotheses[34,67,68], and spatial similarity analysis between encoding models and prefrontal systems[69] or large-scale functional networks[70] (Fig. 1c). (4) Utilize network-based and searchlight-based analysis to compare the predictive performance between local systems and whole-brain level. Based on accumulating evidence for ER deficits in MDs[5,7–11], including excessive cannabis users (CU)[71], we (5) examined the decoders' clinical application potential to detect ER deficits in CU compared to healthy control (HC) participants ($n_{HC}=48$, $n_{CU} = 49$) (Fig. 1d).

Negative feelings

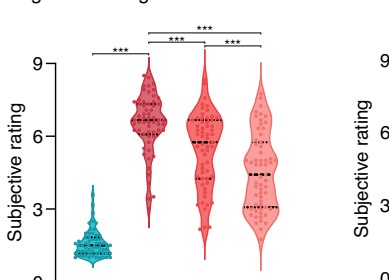
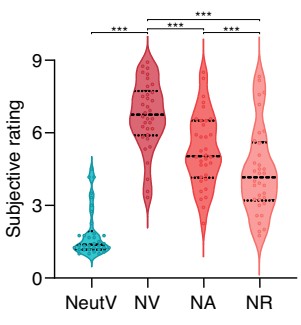

**Fig. 2 | Negative emotional state in the discovery and validation cohort.** Averaged subjective feeling ratings in the discovery (left) and validation cohort (right). Violin plots represent the value distribution; each dot represents the negative rating for individual participants averaged across trials in the corresponding condition. The dotted line in the center represents the median, and the dashed lines on either side indicate the first (Q1) and third (Q3) quartiles. Repeated-measures ANOVA (two-sided) was used to test main effects of stimulus type and strategy, with Greenhouse-Geisser correction applied when Mauchly's test of sphericity was violated. Simple effects analyses further assessed pairwise differences. Source data are provided as a Source Data file. ***$P < 0.001$; exact p values provided in supplements. NV, view (react to) negative clips; NeutV, view (react to) neutral clips; NA, use acceptance strategy to negative clips; NR, use reappraisal strategy to negative clips.

## Results

### Assessment of the naturalistic ER paradigm

Subjective experience and regulation of negative emotions. During fMRI, participants were instructed to react naturally to neutral and negative video clips, and use corresponding ER strategies based on presented cues to negative clips, then report their momentary negative affect on a 9-point Likert scale (9-very negative, 1-not negative at all). In both discovery ($n = 59$) and validation ($n = 33$) cohorts, significant main effects of condition (stimulus type: NV, NeutV; strategies: NV, NA, NR) were observed with repeated-measures analysis of variance (ANOVA), and further simple effects analysis indicated that negative clips induced considerable negative emotions (discovery cohort: $F[1,58] = 1039.434$, $P = 9.96 \times 10^{-39}$, $\eta_p^2 = 0.947$; validation cohort: $F[1,32] = 255.626$, $P = 8.15 \times 10^{-17}$, $\eta_p^2 = 0.947$; Fig. 2, details in supplements). Both strategies effectively attenuate the negative emotions (discovery cohort: $F[2,95] = 81.668$, $P = 3.18 \times 10^{-19}$, $\eta_p^2 = 0.585$; validation cohort: $F[2,52] = 56.637$, $P = 1.32 \times 10^{-12}$, $\eta_p^2 = 0.639$). Reappraisal led to a stronger decrease compared to acceptance (discovery cohort: $P = 2.02 \times 10^{-10}$; validation cohort: $P = 2.4 \times 10^{-5}$). Findings are further corroborated by success rates (discovery cohort: $7.51 \pm 1.01$, validation cohort: $7.53 \pm 0.98$).

### Decoding emotional experience and ER strategies in dynamic naturalistic contexts

Employing the SVM with leave-one-subject-out cross-validation (LOSO-CV) to identify whole-brain fMRI activation predictive signatures of negative emotional experience (NNES), acceptance (NERS-A), and reappraisal (NERS-R) (Fig. 3a–c) during naturalistic emotion processing. Decoders were developed using data from the discovery cohort only. Their performance was validated using cross-validated classification in the discovery cohort and determining the reactivity in an independent validation cohort ($n = 33$) that underwent a similar paradigm. NNES, NERS-A, and NERS-R could accurately discriminate the corresponding verse control condition, respectively (Fig. 3d, Table 1). Testing the signatures without further model fitting in an independent validation cohort revealed robust predictive performances (Fig. 3e).

### Generalization to independent data

To assess the generalizability of the decoders across samples, MRI systems, culture, and stimuli (dynamic versus static or heat-induced pain), we first tested them in a large independent cohort ($n = 358$)[34]. NNES could accurately discriminate exposure to negative versus neutral pictures (Fig. 3f left; Table 1). Consistent with the strategy used by the participants (acceptance or reappraisal)[34], NERS-A and NERS-R could accurately classify the down-regulate vs. experiencing negative emotions. NNES could accurately distinguish between experiencing negative and neutral emotions. In another independent picture-based ER task ($n = 45$)[65], NERS-R reliably classified distancing (a form of reappraisal[72]) vs. experiencing negative emotions, NNES classified experiencing negative vs. neutral emotions (Fig. 3f middle; Table 1). However, NERS-A showed poor performance.

When using the whole-brain data, the decoders exhibited poor predictive performance in distinguishing between reappraisal-associated down-regulating of heat-induced pain and its experience ($n = 33$, Fig. S4c)[66]. One possible explanation refers to the different stimulus modalities (visual-video/somatosensory-heat) engaging distinct sensory processing pathways. After excluding voxels from the visual network[70], the NERS-R could accurately discriminate between reappraising and experiencing heat-induced pain (Fig. 3f right; Table 1). However, NERS-A and NNES still failed to discriminate accurately, underscoring the domain-specificity of the decoders.

Together, these results underscore the robustness and specificity of the decoders for specific mental processes.

### Common and separate signatures of reappraisal, acceptance, and negative emotional experiences

We next systematically evaluated which brain regions/networks provided a consistent and robust contribution to the corresponding mental processes.

Bootstrap tests and encoding model estimation. We initially threshold the multivariate patterns using bootstrapping (Fig. 3a–c). Since decoding model features may capture not only neurobiological processes of interest (e.g., related to ER), but also noise[73], we subsequently transformed the bootstrapped within-subject weighted maps into reconstructed 'activation pattern' (encoding model)[74,75] and estimated them at the population-level (FDR $q < 0.05$, Fig. 4a). This pattern reflects the direction of the relationship between each voxel and the target variable, indicating which voxels are positively or negatively associated with predicted mental processes, e.g., ER strategy. Additional spatial similarity analysis showed that the decoding and encoding models exhibit high spatial similarities across the vast majority of voxels (Fig. S1).

The intense negative emotions induced by watching the negative (versus neutral) clips were accompanied by a positive association with activity in subcortical regions engaged in emotional reactivity, such as the insular, thalamus, periaqueductal gray (PAG), as well as superior parietal (e.g., supramarginal), ventrolateral occipital and posterior mid-temporal regions involved in the integration of information and semantic content[54,76,77]; negative association with activity in a wide-spread bilateral anterior temporal and frontal network, including the pre-supplementary motor area (SMA)[78] and medial/lateral frontal regions.

Acceptance (versus experiencing negative emotion) significantly attenuated activity in subcortical (e.g., fusiform and lingual gyrus), superior and inferior parietal, inferior temporal, and bilateral pars triangularis (IFSa) regions, while concomitantly associated with increased activity in the amygdala (extended to hippocampus), insular cortex, putamen, frontal regions including the prefrontal lobe (bilateral dorsolateral (dlPFC), dorsomedial (dmPFC), ventromedial (vmPFC), ventrolateral (vlPFC) prefrontal cortex, left rostral anterior cingulate cortex (rACC), isthmus cingulate, precentral, lateral occipital

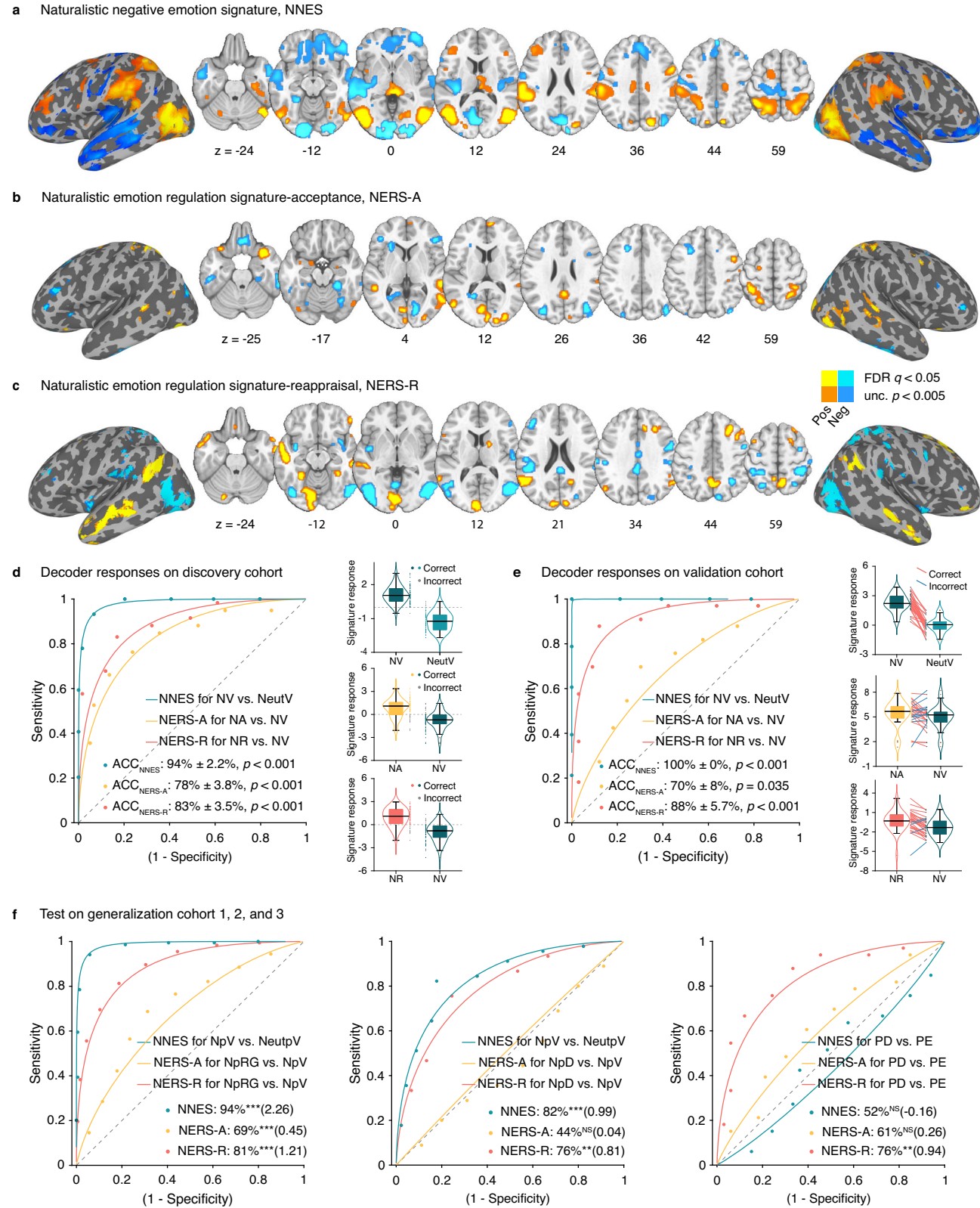

**a**  Naturalistic negative emotion signature, NNES

**b**  Naturalistic emotion regulation signature-acceptance, NERS-A

**c**  Naturalistic emotion regulation signature-reappraisal, NERS-R

**d**  Decoder responses on discovery cohort

**e**  Decoder responses on validation cohort

**f**  Test on generalization cohort 1, 2, and 3

cortex, SMA, and superior parietal regions including the precuneus (Fig. 4b).

Reappraisal (versus experiencing negative emotion) is negatively associated with activity in subcortical regions, including the bilateral amygdala, insula, putamen, and thalamus involved in emotional experience, as well as lateral occipital, posterior frontal, and parietal regions, including the SMA and supramarginal gyrus. Reappraisal is concomitantly associated with activity in a bilateral frontal network, encompassing medial, orbitofrontal, vlPFC, rACC, right dlPFC, as well as parietal regions including the precuneus, superior, and middle temporal regions (Fig. 4c).

The effects were confirmed by the reconstructed activation patterns across datasets and with univariate analyses (Figs. S2 and Fig. S3). Consistent patterns were observed under conditions where the

**Fig. 3 | Developed signatures and predictive performances.** NNES (thresholded at false discovery rate (FDR) $q < 0.05$) (**a**), NERS-A (FDR $q < 0.05$ and unc. $p < 0.005$) (**b**), and NERS-R (FDR $q < 0.05$ and unc. $p < 0.005$) (**c**) pattern maps, based on a 10,000-sample bootstrapping. Hot color indicates positive weights, whereas cold color indicates negative weights. **d** Receiver operating characteristic (ROC) plot illustrates the classification performances of the SVM models on the discovery cohort ($n = 59$). Statistical significance was evaluated using a one-sided binomial test under leave-one-subject-out cross-validation (LOSO-CV) framework, with the misclassification threshold fixed at 0. No multiple-comparison correction was applied because each model was trained and tested once, resulting in only a single statistical test per model. The right three panels show the distributions of the decoders' responses. The boxes are bounded at the first (Q1) and third (Q3) quartiles, with the median shown as the central line. Whiskers extend to the maximum and minimum values within 1.5×interquartile range (IQR) from Q1 and Q3. Data points outside the whisker range are labeled as outliers. The yellow, green, and red dots beside the violin plots indicate correct classification, and the gray dots indicate misclassification. The classification performances of the developed decoders on the validation cohort (**e**, $n = 33$) and generalization cohort (**f**, based on static pictures[34,65], $n = 358$ and $n = 45$, or heat-induced pain[66], $n = 33$) were evaluated using a two-alternative forced-choice procedure and assessed for significance using a one-sided binomial test comparing accuracy against the chance level of 0.5. No multiple-comparison correction was applied. NNES naturalistic negative emotion signature, NERS-A naturalistic emotion regulation signature-acceptance, NERS-R naturalistic emotion regulation signature-reappraisal, NV negative (clips)-view, NeutV, neutral (clips)-view, NA negative (clips)-acceptance, NR negative (clips)-reappraisal, ACC accuracy, NpV negative pictures-view, NeutpV neutral pictures-view, NpRG negative pictures-regulate, NpD distancing negative pictures, PD down-regulate heat-induced pain, PE experience heat-induced pain, NS not significant; **$P < 0.01$; ***$P < 0.001$. Detailed statistical results are provided in Table 1.

## Table 1 | Classification performance on the discovery and validation cohort

| Condition | Decoder | ACC (%) | SD | Sens (%) | CI | Spec (%) | CI | Effect size | *P* |
|---|---|---|---|---|---|---|---|---|---|
| Discovery cohort ($n = 59$) | | | | | | | | | |
| NV vs. NeutV | NNES | 94 | 2.2 | 98 | 95–100 | 90 | 81–97 | 3.01 | **<$1 \times 10^{-50}$** |
| NA vs. NV | NERS-A | 78 | 3.8 | 75 | 68–86 | 81 | 71–91 | 1.37 | **$7.75 \times 10^{-10}$** |
| NR vs. NV | NERS-R | 83 | 3.5 | 83 | 73–92 | 83 | 73–92 | 1.71 | **$1.53 \times 10^{-13}$** |
| Validation cohort ($n = 33$) | | | | | | | | | |
| NV vs. NeutV | NNES | 100 | 0 | 100 | 100–100 | 100 | 100–100 | 3.55 | **$2.33 \times 10^{-10}$** |
| NA vs. NV | NERS-A | 70 | 8 | 70 | 53–86 | 70 | 53–85 | 0.47 | **0.0351** |
| NR vs. NV | NERS-R | 88 | 5.7 | 88 | 75–97 | 88 | 76–97 | 1.42 | **$1.10 \times 10^{-5}$** |
| Generalization cohort 1 ($n = 358$) | | | | | | | | | |
| NpV vs. NeutpV | NNES | 94 | 1.2 | 94 | 92–96 | 94 | 92–96 | 2.26 | **<$1 \times 10^{-50}$** |
| NpRG vs. NpV | NERS-A | 69 | 2.5 | 69 | 64–74 | 69 | 64–74 | 0.45 | **$1.14 \times 10^{-12}$** |
| NpRG vs. NpV | NERS-R | 81 | 2.1 | 81 | 77–85 | 81 | 78–85 | 1.21 | **<$1 \times 10^{-50}$** |
| Generalization cohort 2 ($n = 45$) | | | | | | | | | |
| NpV vs. NeutpV | NNES | 82 | 5.7 | 82 | 70–92 | 82 | 70–93 | 0.99 | **$1.54 \times 10^{-5}$** |
| NpD vs. NpV | NERS-A | 44 | 7.4 | 44 | 29–60 | 44 | 30–60 | 0.04 | $5.51 \times 10^{-1}$ |
| NpD vs. NpV | NERS-R | 76 | 6.4 | 76 | 62–87 | 76 | 64–87 | 0.81 | **$8.24 \times 10^{-4}$** |
| Generalization cohort 3 ($n = 33$) | | | | | | | | | |
| PD vs. PE | NNES | 52 | 8.7 | 52 | 35–69 | 52 | 33–69 | −0.16 | 1 |
| PD vs. PE | NERS-A | 61 | 8.5 | 61 | 44–77 | 61 | 44–77 | 0.26 | 0.2962 |
| PD vs. PE | NERS-R | 76 | 7.5 | 76 | 60–90 | 76 | 60–90 | 0.94 | **0.0045** |

For statistics from the discovery cohort, classification significance was assessed using one-sided binomial tests under a leave-one-subject-out cross-validation (LOSO-CV) framework. For the validation and generalization cohorts, performance was evaluated using a two-alternative forced-choice procedure and significance was assessed using a one-sided binomial test comparing accuracy against the chance level of 0.5. No multiple-comparison correction was applied. Bold indicates $P < 0.05$.

ACC, accuracy; Sens, sensitivity; specificity; SD, standard deviation; CI, confidence interval; NV, view (react to) negative clips; NeutV, view (react to) neutral clips; NA, use acceptance strategy to negative clips; NR, use reappraisal strategy to negative clips; NNES, naturalistic negative emotion signature; NERS-A, naturalistic emotion regulation signature – acceptance; NERS-R, naturalistic emotion regulation signature – reappraisal; NpV, view negative picture; NeutpV, view neutral picture; NpRG, negative picture-regulate; NpD, distancing negative pictures; PD, down-regulate heat-induced pain; PE, experience heat-induced pain.

decoders achieved significantly accurate predictions (e.g., NERS-R across all datasets). Some inconsistencies were observed in generalization cohorts 2 and 3 for NERS-A, reflecting lower predictive accuracy when participants performed a different strategy (reappraisal rather than acceptance). These results demonstrate that the decoders reliably capture the neural representations associated with the specific mental processes they were trained to predict.

Brain systems identified by BF analysis. BF analysis was used to determine the common and separate neuro-basis between acceptance and reappraisal. 'Acceptance only' brain regions were identified by activation during acceptance (BF > 5) but not for reappraisal (BF < 1/5); reverse for 'Reappraisal only' (reappraisal BF > 5, acceptance BF < 1/5); 'Common' brain regions were identified by activation during both acceptance (BF > 5) and reappraisal (BF > 5)[34,67].

Effective ER via both strategies robustly recruited cortical midline DMN regions including the bilateral dmPFC (extending to rACC), right inferior precuneus, left cuneus, middle temporal and lateral occipital regions (Fig. 5a, conjunction analysis in Fig. S4). Acceptance and reappraisal specifically recruited distributed process-specific systems spanning several cortical systems, with the acceptance primarily engaging somatomotor (and to a lesser extend ventral attention and frontoparietal) large-scale networks and subcortical systems (e.g., superior temporal lobe, dlPFC, vlPFC, insular, amygdala, and putamen), while reappraisal primarily engaged an extensive the frontoparietal and DMN regions (e.g., superior frontal and inferior parietal regions, dlPFC, vlPFC, supramarginal, and precuneus).

Spatial similarity between stable decoding maps and interest networks/crucial ROIs. To provide a more quantitative measure of the

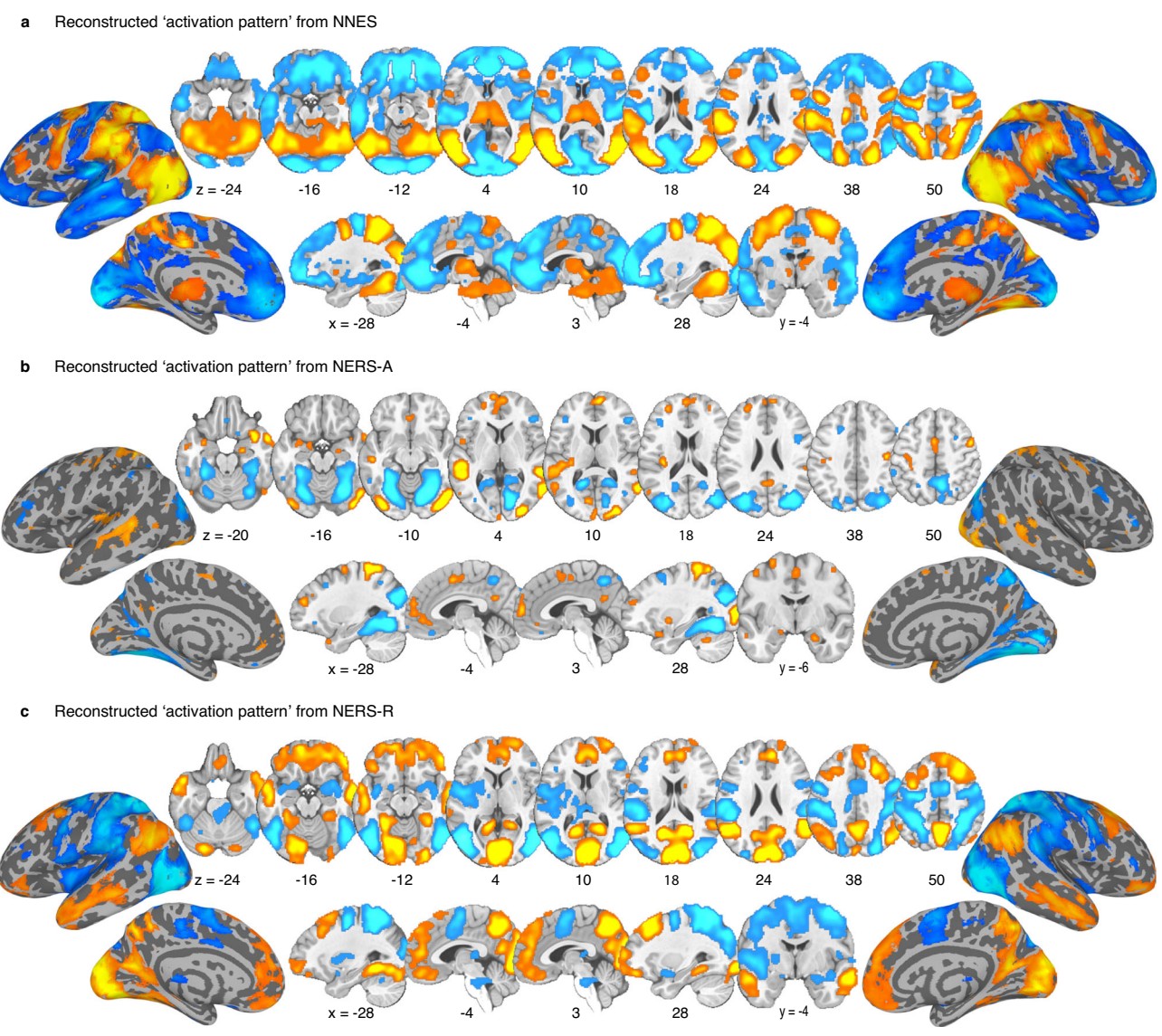

**Fig. 4 | Reconstructed 'activation pattern'.** FDR corrected ($q < 0.05$) group-level reconstructed 'activation pattern' transformed from NNES (**a**), NERS-A (**b**), and NERS-R (**c**). The color indicates the direction of the relationship between each voxel and the target variable – i.e., which voxels are positively (red) or negatively (blue) related to the corresponding mental processing (e.g., ER strategies). The hot color indicates positive associations, whereas the cold color indicates negative associations.

extent to which specific prefrontal anatomical regions[69] or large-scale networks[70] are consistently involved in emotional experience and regulation, we estimated the spatial similarities between the reconstructed 'activation pattern' and anatomical and functional templates of frontal regions or large-scale networks, respectively (Fig. 5b, c). For each region, we calculated the ratio of contribution for acceptance versus reappraisal versus experience negative affect (pie plot in Fig. 5b, c; Table S1). The contribution is defined as the percentage of the region encoded by each model[53,79], while also directly comparing acceptance and reappraisal (Fig. 5d). Except the vlPFC, which was engaged in all processes, prefrontal regions exhibited a stronger engagement during both regulation strategies as compared to experiencing the emotions. While the vmPFC, dmPFC, and vmPFC showed comparable contributions to both strategies, the ACC had stronger engagement for acceptance, whereas the dlPFC and orbitofrontal cortex (OFC) exhibited stronger engagement during reappraisal. On the network level, the visual networks exhibited equal contribution across the three models; the experience of negative emotion may be mostly encoded in the dorsal, ventral attention, and somatomotor networks. When comparing ER strategies, the somatomotor and attention networks

presented higher contributions for acceptance, while the frontoparietal networks had relatively stronger contributions for reappraisal, and the DMN exhibited similar contributions for both strategies.

To provide a more precise, quantitative measure of how similar or different the acceptance and reappraisal are encoded across the entire brain, we additionally analyzed the voxel-level spatial covariation between the unthresholded normalized (z-scored) encoding maps of NERS-A and NERS-R[80]. Figure 5e illustrates the joint distribution of the voxel weights of the encoding model of NERS-A and NERS-R. Octants with different colors indicate voxel weights of shared positive/negative, selective positive/negative weights, and opposite for the two models. A high proportion of weights (sum of squared distances to the origin) clustered in the nonshared octants (5,7) and opposite octants (4,8), while a comparably small proportion of weights clustered in the shared octant 6, further supporting that separable brain representations contribute to implementing acceptance and reappraisal strategies. This was further confirmed in the results of the decoding maps (Fig. S4d).

Predictive performances between decoders. To determine whether our decoders reflected both the common and separate

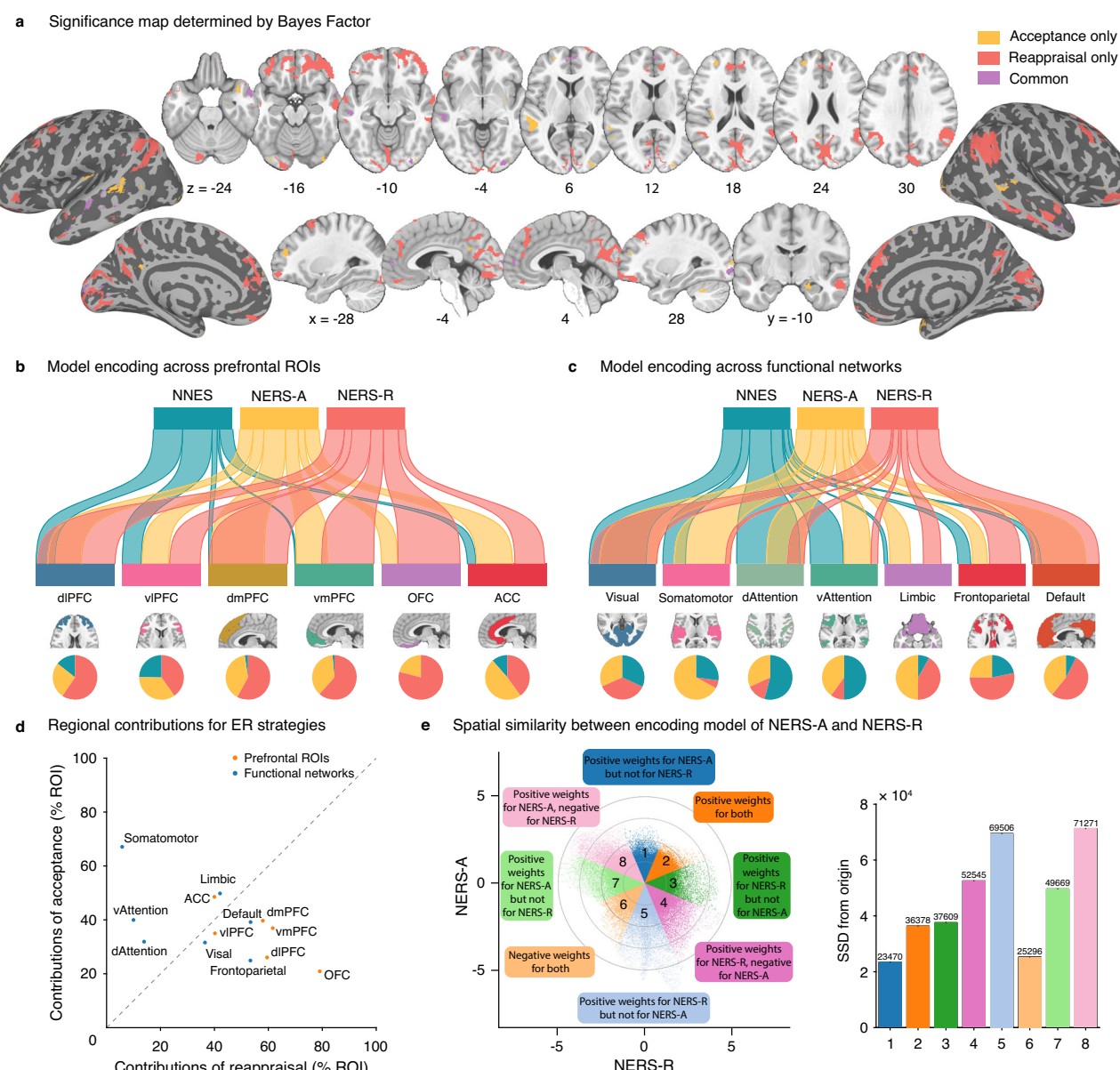

**Fig. 5 | Comparing acceptance, reappraisal, and negative emotion signatures. a** Brain regions of the class of each voxel (Acceptance only, Reappraisal only, or Common). River plots illustrated the spatial similarity (cosine similarity) between reconstructed 'activation pattern' and anatomical parcellation of the prefrontal cortex[69] (**b**), or resting-state-based functional parcellation of cortical regions[70] (**c**). Reconstructed 'activation patterns' were FDR-thresholded ($q < 0.05$), and only positive voxels were included in similarity analyses. In **b**, **c**, the thickness of the ribbons indicates the normalized maximum cosine between sets. The pie charts indicate the relative contributions of each pattern to each ROI (**a**) or network (**b**) (i.e., the percentage of voxels with the highest cosine similarity for each map). **d** Regional contribution scores for acceptance versus reappraisal[79]. X-axis—reappraisal model contribution (percentage of ROI or network occupied by the encoding model of reappraisal), y-axis—acceptance model contribution (percentage of ROI or network occupied by the encoding model of acceptance). The gray line represents equal contribution in both models; a greater distance from the line indicates stronger imbalance. Points on the left indicate higher contribution

for acceptance, whereas points on the right indicate higher contribution for reappraisal. **e** Voxel-level spatial similarity between the encoding model of NERS-A and NERS-R. Scatter plots illustrate normalized unthresholded voxel beta weights of the reconstructed 'activation pattern' of NERS-A (y-axis) and NERS-R (x-axis). Colored octants indicate voxels of shared positive (Octant 2) or shared negative (Octant 6), selective positive weights for NERS-A (Octant 1) or NERS-R (Octant 3), selective negative weights for NERS-A (Octant 5) or NERS-R (Octant 7), and voxel weights opposite for the two signatures (Octants 4 and 8). The right bars indicate the sum of squared distances from the origin (0, 0) for each octant, which integrates the number of voxels and combined weights. The numerical values shown at the top of each bar represent these total squared distances for the corresponding octant. See Fig. S4d for the results of voxel-level spatial similarity between the decoding model of NERS-A and NERS-R. dlPFC dorsolateral prefrontal cortex, vlPFC ventrolateral prefrontal cortex, dmPFC dorsomedial prefrontal cortex, vmPFC ventromedial prefrontal cortex, OFC orbitofrontal cortex, ACC anterior cingulate cortex, dAttention dorsal attention, vAttention ventral attention.

processing of naturalistic acceptance, reappraisal, and negative emotion, we also compared the decoders' performances on different conditions from the validation cohort and added a neural signature for negative affect (PINES, picture-induced negative emotion signature[81]). Except for NERS-R, the other three decoders could distinguish between NV and NeutV, but NNES has the highest accuracy

and effect size (Fig. S5 and Table S2). NNES and PINES show low performance when predicting ER processing (acceptance and reappraisal). Only NERS-A could accurately classify between NA and NV. Both NERS-R and NERS-A could classify between NR and NV, but NERS-R is more accurate with a larger effect size. Together, these findings indicate that our decoders captured and further

Network-based predictions

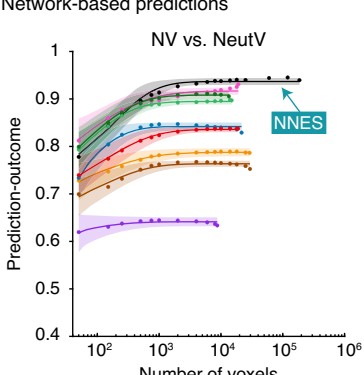 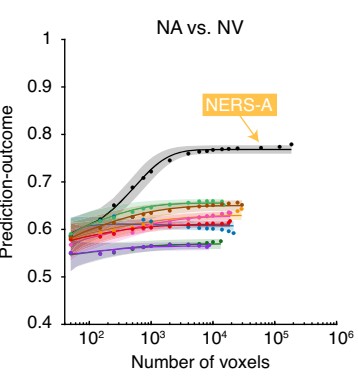 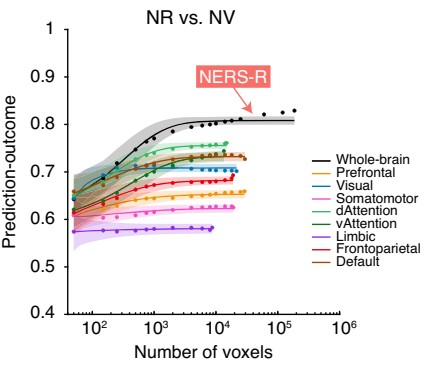

**Fig. 6 | Predictive performance: network-based prediction.** Regulating or experiencing naturalistic negative emotions is distributed across multiple systems. The model performance was evaluated as increasing numbers of voxels/features (x-axis) were selected to predict experience naturalistic negative emotions (NV vs. NeutV), use acceptance (NA vs. NV), and reappraisal (NR vs. NV) strategy in different ROIs including the whole brain (black), prefrontal cortex (light orange), or individual large-scale cerebral networks (other colored lines). The y-axis denotes the cross-validated classification accuracy. Colored dots indicate the mean correlation coefficients, solid lines indicate the mean parametric fit, and shaded regions indicate the standard deviation.

demonstrated the common and separate nature of the above three mental processes.

### Evaluation of predictive performances based on whole-brain or local systems

Given the continuing debate about the specific systems underlying ER, e.g., the dlPFC[33,34,82], DMN[39,83,84], and emerging evidence that emphasizes the distributed neural computations at whole-brain-level[52-55], we employed network-based and searchlight-based analyses to estimate the predictive performances of local brain regions/networks and compared their performances with whole-brain decoders. To control for the potential effects of the number of voxels/features in prediction (i.e., the whole-brain model contains many more features), voxels were randomly selected (repeated 1000 times) from a uniform distribution spanning the whole brain (Fig. 6, black), prefrontal cortex (light orange, which has been suggested play crucial roles in both generate and regulate emotions[32,33,36]), or individual large-scale functional networks (averaged over 1000 iterations)[53-55]. The asymptotic prediction when sampling from the whole-brain, as we trained decoders (black line in Fig. 6), was more accurate than the asymptotic prediction within individual networks (colored lines), especially with more available voxels (details in Supplementary). Moreover, model performance was optimized (i.e., reaching asymptote) when approximately 10,000 voxels were randomly sampled across the whole brain, further confirming that information about ER processing is contained in patterns of activity that span multiple systems. The above results can also be demonstrated by searchlight analysis, although statistically significant and thresholded brain regions are similar to the whole-brain decoders (Fig. S6), the effect sizes in terms of prediction-outcome accuracy were substantially smaller than those obtained from the developed decoders ($P < 0.001$). Together, findings underscore that ER and negative emotional experiences are encoded in distributed neural patterns that span multiple systems, rather than single brain regions or networks.

### Translation into a neuromarker for determining ER deficits in MDs

To test whether our decoders can capture deficient ER processing in individuals with MDs[10,71,85], we applied the decoders to extended data from our previous study[71] describing ER deficits in CU. We incorporated additional unpublished data from individuals with CU disorder and matched controls who underwent a similar picture-based reappraisal paradigm with concomitant fMRI[71].

A total of 48 HC and 49 CU participants were included. Consistent with the initial study, behavioral analyses confirmed ER deficits in the CU (reappraisal success[71], $0.46 \pm 0.46$) as compared to HC ($0.78 \pm 0.54$) ($t = -3.072$, $P = 0.003$, Cohen's d = 0.50).

NERS-R could accurately predict neural activity during reappraisal (distancing) (NpD) vs. viewing (NpV) negative pictures in HC but not in CU participants (Fig. 7, Table 2), and exhibited marginally significant differences between groups in terms of the (NpD-NpV) pattern expression (accuracy=$61 \pm 5\%$, $P = 0.053$)[56]. NNES could accurately distinguish between NpD and view neutral pictures (NeupV) in both groups, underscoring impaired regulation rather than excessive emotion reactivity as a neurofunctional marker for CU. Further permutation tests revealed that the pattern expression of the two groups has significant differences for NERS-R and NNES (both $P < 0.0001$), which are consistent with the reconstructed "activation patterns". In contrast, NERS-A showed poor predictive performance.

## Discussion

Identifying comprehensive and accurate brain models for the common and distinct brain representations that underlie ER is critical for advancing mechanistic models of cognitive regulation as well as for developing accurate biomarkers for regulatory dysfunction in MDs. While reappraisal and acceptance are widely implemented in therapeutic contexts and have been debated from theoretical[1,3], experimental[34], and clinical[14,86] perspectives, the underlying common and distinct brain processes in ecologically valid dynamic contexts have not been accurately described.

Here, we combined immersive naturalistic neuroimaging with predictive modeling to systematically establish and validate three comprehensive, accurate, and sensitive brain-based models for negative affect (NNES) and its regulation via acceptance (NERS-A) and reappraisal (NERS-R). The developed decoders are generalizable across imaging systems, cultures, and paradigms. Next, through systematic analyses, we uncovered the shared and separable neural representations of acceptance, reappraisal, and negative emotion, which indicated that these mental processes are encoded in distributed and distinguishable neural systems on the whole-brain level, yet also show process-specific engagement of large-scale systems. Both acceptance and reappraisal engaged cortical midline regions of the core DMN, including the dmPFC and posterior parietal cortex, indicating a general contribution of this system to effective ER. Each strategy, furthermore, exhibited distinct profiles aligned with its cognitive demands. Acceptance was encoded in distributed regions

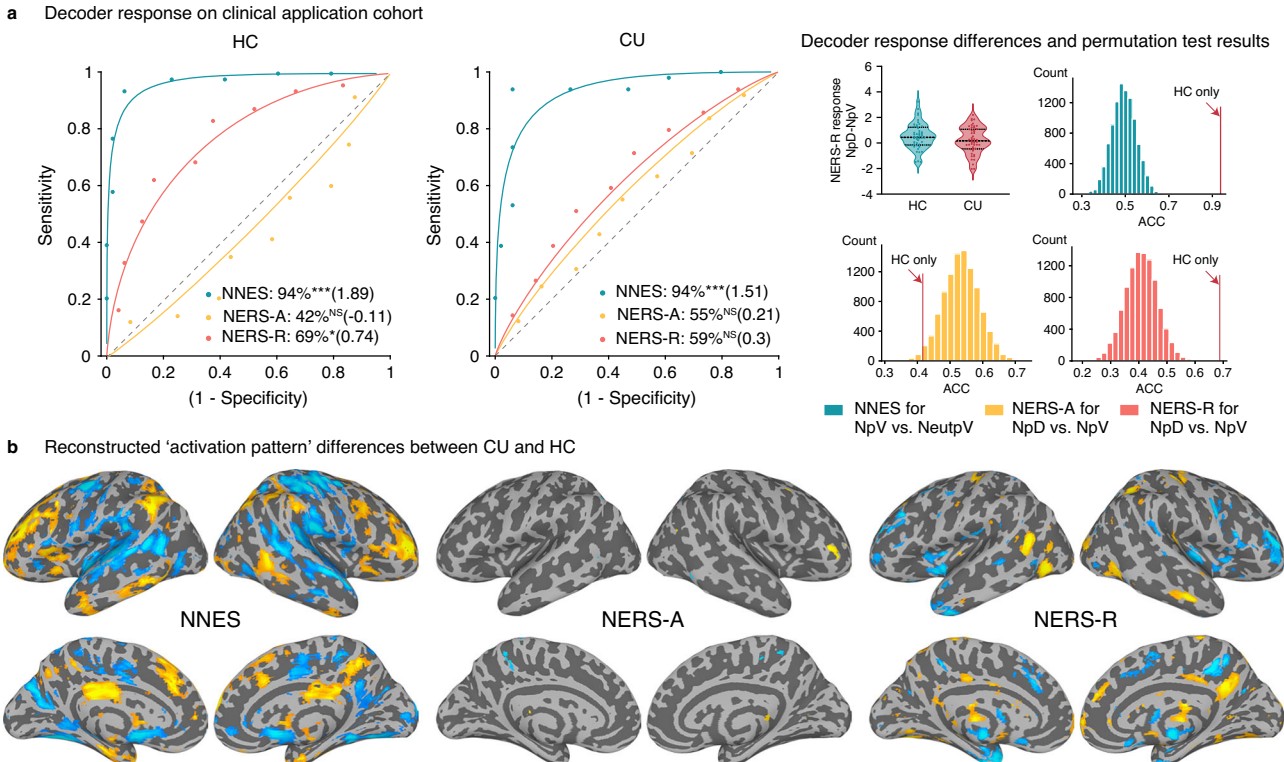

**a** Decoder response on clinical application cohort

Decoder response differences and permutation test results

NNES for NpV vs. NeutpV — NERS-A for NpD vs. NpV — NERS-R for NpD vs. NpV

**b** Reconstructed 'activation pattern' differences between CU and HC

NNES    NERS-A    NERS-R

**Fig. 7 | Testing the clinical application potential of the neuromarkers. a** The NERS-R could accurately distinguish between distancing (NpD, another form of reappraisal) and viewing (NpV) negative pictures in health control (HC, n = 48) participants, but not for cannabis users (CU, n = 49)[71], with marginally significant intergroup differences in (NpD-NpV) pattern expression (accuracy=61 ± 5%, P = 0.053)[56]. Violin plots represent the differences in NERS-R responses to NpD minus NpV, in HC versus CU. Permutation tests reveal that the pattern expression of the two groups has significant differences (red bar chart). NNES could accurately distinguish between NpV and view neutral pictures (NeutpV); the pattern expressions of the two groups are also significantly different (green bar chart). **b** Brain regions showing significant differences in the reconstructed activation pattern when comparing CU and HC participants (FDR q < 0.05). *P < 0.05, ***P < 0.001, NS not significant (binomial test, two-sided, uncorrected).

spanning subcortical, somatomotor, and ventral attention networks, consistent with embodied awareness and experiential processing. Reappraisal, by contrast, relied on regions spanning the fronto-parietal control network, including lateral frontal and inferior parietal regions, reflecting its reliance on executive functions and semantic updating. The robustness of these observations is further supported by the consistent reconstructed brain activation patterns across multiple datasets and traditional univariate analyses. Of note, the NERS-A, trained on experimental data using the acceptance strategy, did not generalize to experimental data obtained during emotion regulation via reappraisal (generalization cohorts 2 and 3), which underscores – to a certain extent – the process-specificity of the neural decoders. While the distinction was further substantiated by BF analyses and spatial similarity mapping, decoding performance was substantially higher for whole-brain decoders as compared to regional models, underscoring the importance of brain-wide integration for capturing emotional experience and regulation. Finally, the neurofunctional decoders could detect reappraisal-specific ER deficits in a large set of CU, underscoring the sensitivity of the signatures to detect specific behavioral impairments on the neural level and clinical application potential.

Combining naturalistic fMRI with mass-univariate and multivariate predictive analyses, we established comprehensive and distributed brain models for negative emotional experiences and their cognitive regulation in dynamically close-to-real-life contexts. The affective videos induced immersive and strongly negative emotional states, which were accompanied by positive engagement of brain regions involved in emotional experience, physiological and affective arousal, including the insula, PAG, thalamus[32,33,77,79,87], and negatively

associated with the engagement of a widespread frontal and temporal network involved in cognitive and regulatory control. Spatial similarity and network-level analyses further confirmed the contribution of large-scale cortical networks, including the attention and visual networks (resembling previous studies[88] using dynamic stimuli). However, both the developed (NNES) and an established signature for negative affect (PINES) poorly predict the ER strategies, complemented by dissociable neural responses, most notably between experience and reappraisal of negative emotions across multiple brain regions, supporting previous research indicating that emotion generation and emotion regulation can be distinguished at the level of distributed neural representations, despite substantial overlap in the underlying neural systems[31–33]. The relatively poor performance of PINES when predicting data based on naturalistic dynamic stimuli may also underscore that the neural systems for processing emotional information in a dynamic context may differ from those engaged by sparsely presented stimuli in conventional laboratory experiments (see ref.[27,50,51]). Although our results indicate differences between static and naturalistic paradigms, we do not claim that naturalistic stimuli are inherently superior, as multiple factors may contribute to these observed differences.

Both ER strategies could efficiently reduce negative emotional states, indicating that utilizing reappraisal and acceptance represents an efficient means to regulate negative emotions and the associated neural systems in dynamic, ecologically-valid contexts. Consistent with ref.[24], reappraisal led to a greater reduction of subjective negativity compared to acceptance.

On the neural level, ER via both strategies was predicted by neural regulation in distributed cortical and subcortical systems, with

**Table 2 | Classification performance on the clinical application cohort**

| Condition | Decoder | ACC (%) | | Sens (%) | | Spec (%) | | Effect | P |
|---|---|---|---|---|---|---|---|---|---|
| | | | SD | | CI | | CI | size | |
| Clinical application cohort | | | | | | | | | |
| HC (n = 48) | | | | | | | | | |
| NpV vs. NeutpV | NNES | 94 | 3.5 | 94 | 86–100 | 94 | 86–100 | 1.89 | **$1.31 \times 10^{-10}$** |
| NpD vs. NpV | NERS-A | 42 | 7.1 | 42 | 27–56 | 42 | 29–55 | −0.11 | 0.3123 |
| NpD vs. NpV | NERS-R | 69 | 6.7 | 69 | 56–82 | 49 | 56–82 | 0.74 | **0.0133** |
| CU (n = 49) | | | | | | | | | |
| NpV vs. NeutpV | NNES | 94 | 3.4 | 94 | 87–100 | 94 | 87–100 | 1.51 | **$6.98 \times 10^{-11}$** |
| NpD vs. NpV | NERS-A | 55 | 7.1 | 55 | 41–69 | 55 | 40–69 | 0.21 | 0.5681 |
| NpD vs. NpV | NERS-R | 59 | 7 | 59 | 46–73 | 59 | 46–73 | 0.3 | 0.2528 |

Statistical performance was evaluated using a two-alternative forced-choice procedure, and significance was assessed using a one-sided binomial test comparing accuracy against the chance level of 0.5. No multiple-comparison correction was applied. Bold indicates $P < 0.05$.
*HC* healthy control, *CU* cannabis users, *NpV* view negative pictures, *NeutpV* view neutral pictures, *NpD* distancing negative pictures.

common and distinct signatures. BF and conjunction analyses demonstrated that effective ER by both strategies engaged core regions of the DMN that are strongly involved in the evaluation of the mental state of oneself and others (dmPFC)[21,32,36], episodic auto-biographical memory (inferior precuneus)[89,90], rapid emotional conflict adaptation (rACC)[91], self-awareness and interpersonal processing (temporal regions)[92,93]. Given that both ER strategies require the appraisal of the emotional state concerning the self, the dmPFC and other core DMN regions may support appraisal, metacognitive evaluation, and internal self-reference, enabling ER based on internal goals[94]. On the anatomical level, these functions of the DMN, in particular the dmPFC, may be supported by interconnections with the rostromedial prefrontal, lateral orbitofrontal (lOFC), and pregenual ACC, which may support its role as a central hub for the implementation of both ER strategies[36,94]. The effective regulation was accompanied by reduced engagement of a widespread bilateral network including bilateral superior parietal, inferior and precentral frontal, and lateral occipital regions, which may reflect attenuated motor readiness and expressive output, as well as attenuated visual processing of emotionally salient stimuli, and in turn may promote regulatory control across both strategies[95–97].

Distinguishable contributions were observed in somatomotor[98], attention, or the frontoparietal control network[84,99], respectively. In line with the notion that reappraisal represents an active and effortful ER strategy[23,24], effective implementation specifically engaged an extensive bilateral network of distributed regions within the fronto-parietal control network. These included lateral frontal regions involved in working memory (dlPFC)[82], general and valence-specific inhibition (vlPFC, lOFC)[32,36,100,101], and medial prefrontal regions involved in computation of value or safety (vmPFC, mOFC)[36,102–104]. Parietal regions were specifically recruited during reappraisal, including regions involved in self-referential processing (superior precuneus)[105–107], attentional control (inferior precuneus)[108], and conscious representation of affective semantic content[109]. In contrast, reappraisal specifically decreased engagement of brain regions traditionally linked to emotional "reactivity", including subcortical systems such as the amygdala[32,33,102,108,110–113] and the anterior insular cortex[102]. Interestingly, the SMA, which has been reported to be positively associated with reappraisal success[30], showed decreased engagement during reappraisal in the present study. Possible explanations may include the use of dynamic and relatively long-duration naturalistic stimuli and the associated higher ecological validity of our study.

Acceptance was also encoded in some distinguishable but rather focal prefrontal regions (e.g., dlPFC), yet showed stronger engagement of distributed regions spanning the somatomotor (e.g., superior parietal lobe, SMA, posterior insula)[114,115] and ventral attention networks (e.g., preSMA). Compared to reappraisal, enhanced engagement of

these regions may reflect that effective acceptance relies on goal-oriented attentional deployment (superior parietal lobe[11,116,117]), inter-oception (posterior insula[118,119]), and embodied cognition-affect processing (SMA/preSMA[43]). Notably, acceptance is specifically negatively associated with the brain systems involved in self-referential processing (superior precuneus)[105–107] and threat perception (fusiform)[120,121], while it was positively associated with activation in subcortical systems traditionally linked to emotional reactivity, including the amygdala, hippocampus, and putamen. The stronger engagement of these systems may reflect the non-judgmental awareness of affective arousing and sensory experience and may promote hippocampus-dependent encoding and contextualization[122], in turn facilitating adaptive integration of the emotional experience with autobiographical and situational contexts[24,122–125].

The present findings highlight the fundamental principle that complex cognitive functions like ER are implemented by brain-wide, distributed systems. This is evidenced by the consistently superior classification accuracy of our whole-brain decoders compared to region-specific models, a result further confirmed through both network-based and searchlight-based analyses. The importance of individual brain regions or networks in the experience and regulation of negative emotions has been emphasized in previous frameworks[31–33]. While our findings support preferential engagement of specific networks by distinct strategies – the fronto-parietal control network during reappraisal and the somatomotor network during acceptance – our large-scale network- and searchlight-based predictive analyses further highlight that, under dynamic and naturalistic conditions, both strategies rely on the coordinated activity of distributed and distinguishable brain-wide representations, which is in line with previous findings regarding other mental processes, e.g., digust[53], fear[54], or pain[55]. The distributed engagement was further supported by a system-specific analysis of the prefrontal cortex, commonly identified as systems promoting distinct ER strategies[31–33]. While most prefrontal systems contributed to both strategies, the orbitofrontal cortex showed a markedly higher engagement during reappraisal, which may be due to its extensive anatomical connections and functional coupling with the amygdala during reappraisal, and which is underscored as this region is critical for mediating effective reappraisal of negative emotions[96,126–128]. In general, our findings align with a core principle of modern neuroscience: that sophisticated cognition and affect, including ER, are best characterized by distributed neural computations that integrate information across the entire brain[52–55].

Of note, our findings do not rely on a strict top–down or strong neural separability account of emotion regulation. Recent theoretical perspectives have emphasized substantial neural overlap between emotion generation and emotion regulation, proposing that both

processes draw on shared valuation-related systems while preferentially engaging distinct subprocesses depending on contextual demands and regulatory goals[3,27,129]. Interpreted within this perspective, the present results suggest that in the context of emotion regulation, acceptance and reappraisal recruit common, distributed neural systems, in particular core regions of the DMN, while also recruiting distinguishable engagement of frontoparietal, somatomotor, attentional, and subcortical systems. The observed distinguishability may, to a certain extent, reflect differences in how shared neural resources are coordinated and weighted, rather than the engagement of entirely separate neural modules.

Importantly, exploratory analyses further suggested that our neurofunctional signatures for ER may hold clinical translational potential in terms of determining process-specific alterations in mental disorders and individual-level assessment of neurofunctional alterations. The decoders captured ER strategy-specific neural representation of impaired reappraisal success in CU on the group level, offering neurofunctional evidence for ER-specific deficits in this population[71]. The specific behavioral deficits in terms of reappraisal deficits were reflected on the neural level, such that the NERS-R accurately predicted reappraisal in HC but failed to distinguish reappraisal from negative experience in CU. Importantly, both permutation tests and reconstructed activation patterns convergently revealed that NERS-R and NNES can accurately track abnormal brain activation patterns in CU. Impaired ER and altered emotion reactivity have been widely demonstrated across multiple MDs, including CU[71,130,131], opioid use disorders[85,132,133], alcohol use disorders[134,135], cocaine use disorder[136,137], major depressive disorder[138], anxiety disorders[139,140], and have been mainly related to alterations in regional activity, in particular deficient prefrontal engagement or amygdala regulation. However, precise neuromarkers for tracking emotional dysfunction at the neurofunctional level have not been developed or validated (for convergent efforts see ref.[56,141–143]). Given the critical and transdiagnostic role of ER deficits in MDs[5,7–11], a recent review[62] suggests that machine-learning-based algorithms could identify neuromarkers that provide more precise insights into the underlying pathophysiological processes and inform conventional diagnostic and treatment approaches. Within this context, the developed multivariate predictive decoders may further offer the opportunity to determine individual-level neuromarkers (pattern expressions) to quantify ER ability (e.g., reappraisal) at the neurofunctional level[56]. Moreover, our markers, which captured domain-specific abnormalities in CU participants, may serve as brain-based treatment-response markers for behavioral and brain-based intervention[112,144], and allow for profiling process-specific modulatory effects of novel pharmacological interventions under close to real-world contexts[145,146]. Future studies are required to validate the accuracy, clinical utility, and sensitivity of those decoders across different populations and MDs.

Limitations of the present study are inherent to technical limitations. While combining fMRI with MVPA gives a relatively precise way to investigate brain activity, and has been revealed to be consistent with the conventional univariate approach, limitations are inherent to fMRI (e.g., indirectly measuring neuronal activity, 3 T fMRI may not be able to precisely localize to the midbrain and brainstem nuclei[79]). Although subjective reports of emotional responses have been widely used in previous studies and the neural signatures of subjective experiences and autonomous responses are (partly) separable (e.g., ref.[147–149]), the simultaneous acquisition of physiological measures (e.g., skin conductance, eye tracking) may provide additional information. While previous studies have demonstrated the advantages of naturalistic paradigms in terms of ecological validity[46–51,63,64], further research directly comparing reappraisal and acceptance processes under naturalistic paradigms versus static-image-based paradigms are required to directly compare the neural processes underlying different emotion regulation processes in response to static (e.g., pictures) and dynamic (e.g., movies) stimuli.

In conclusion, we developed and validated three robust, generalizable, and clinically translatable whole-brain neural signatures for negative emotional experience and its effective regulatory control via acceptance and reappraisal in dynamic naturalistic contexts. Our within-subject design enabled a systematic comparison of the common and distinct neural bases underlying these ER strategies and suggests that core DMN regions mediate effective regulation across strategies. On the whole brain level, the strategies were distinguishable, and the predictive neurofunctional signatures demonstrated high robustness across independent cohorts, experimental paradigms, and MRI systems. The findings emphasize that an effective ER is encoded not in isolated brain regions but in synergistic, distributed whole-brain activity patterns.

Application to individuals with ER deficits underscored that the neural decoders offer substantial translational potential. Future applications potentially include: (a) serving as neural markers to assess intervention effects targeting acceptance and reappraisal; (b) characterizing ER-specific neural impairments across MDs; (c) delineating common and distinct neural architectures of various ER strategies; and (d) informing frameworks centered on the role of the DMN or on hierarchical neurocognitive models in ER[150,151]. Together, this work provides a foundation for developing mechanistically-informed, brain-based targets and outcome measures for clinical interventions aimed at enhancing emotional resilience.

## Methods

### Participants

Discovery cohort. Sixty-eight healthy participants were recruited from local universities. Exclusion criteria included left-handedness, color blindness or weakness, and current physical or mental illnesses. Six participants were excluded due to excessive head motion ( > 3 mm or 3°), and three due to failed post-check. Fifty-nine participants (19 male, $21.3 \pm 1.7$ years old, mea$n \pm$ SD) were included in the final analysis.

Validation cohort. Thirty-four healthy participants were recruited as the validation cohort (11 male, $20.3 \pm 1.4$ years old), with the same exclusion criteria as the discovery cohort. One participant was excluded due to being detected as an outlier by the fmri_data.mahal function in the Canlabcore toolbox ($P < 0.05$, Bonferroni corrected).

All participants provided written informed consent. All participants were compensated 140 RMB after the experiment. This pre-registered study (discovery cohort: https://osf.io/s3bp2, validation cohort: https://osf.io/68jkw) was approved by the ethics committee of the University of Electronic Science and Technology of China and in accordance with the Declaration of Helsinki.

### Stimuli and paradigm

Discovery cohort. A total of forty-eight silent video clips were used in the current ER paradigm, rated by one presented once by E-prime software (Version 3.0; Psychology Software Tools, Sharpsburg, PA). Twelve neutral clips include normal driving records, surveillance, vlogs, etc.; thirty-six negative clips include threatening animals, human fights, bullying, car accidents, horror movies, etc. Neutral and negative clips are matched by basic scenarios and characters.

The experiments consisted of four conditions: react (view)-neutral (NeutV), react-negative (NV), reappraise-negative (NR), and accept-negative (NA), each with twelve clips, divided into four runs. To alleviate the potential cognitive overload caused by switching mindsets too often or the potential fatigue or boredom caused by using the same mindset for a long period, we combined "react" with "reappraisal" or "react" with "accept" in each run, and used the ABBA method between participants to balance the order effect (see Fig. 1a).

Participants were provided with written instructions and trained to use a certain mental strategy ("react", "reappraisal", or "accept")

during the subsequent presentation of the video clip. Participants received ~20 minutes of training before scanning. The instructions and procedures were closely aligned with evaluated procedures from previous studies on emotion regulation via reappraisal and acceptance (e.g., ref.[23,26]). For "react", participants were asked to react and feel naturally, and not try to mitigate their reaction in any way. For "reappraisal", participants were asked to regulate their emotion via re-interpreting the content and context of the video, for instance, via imagining that there would be a positive outcome. For "accept", participants were asked to experience their feelings within an "inaction" framework and not to try to judge, control, change, or fight against the emotions. Each trial began with a fixation cross (3–5 s), followed by a 2 s cue, then a fixation cross last 6–8 s, after that a 25 s video clip, followed by a fixation cross (1–1.5 s), then a 9-point Likert scale (4 s) for participants to report "How negative do you feel right now?" (9–very negative, 1–not negative at all). At the end of each run, participants will rate "On average, how successfully have you used the indicated mindset in this run?" (9–very successful, 1–not successful at all).

Validation cohort. The stimuli, experiment conditions, and trial structure are identical to the discovery cohort. However, to control the potential influence of the "ER trial" on the "react trial" in the same run, we placed all 12 trials of each condition within one run (randomly presented), using the Latin square design to balance order effects between participants.

## MRI data acquisition and preprocessing
Discovery cohort. MRI data were collected on a 3.0 T scanner (Ingenia Elition; Philips Healthcare, Best, Netherlands) with a 32-channel head coil. Structural images were acquired using high-resolution three-dimensional T1-weighted images (repetition time (TR) = 8.2 ms, echo time (TE) = 3.8 ms, field of view (FOV) = 243 mm, flip angle (FA) = 8°, 243 × 243 matrix, 1 × 1 × 1 mm voxels, 180 sagittal slices, phase encoding anterior » posterior) and were used for anatomical localization and warping to the standard Montreal Neurological Institute (MNI) space only. Functional images were acquired using a gradient-recalled echo (GRE) echo-planar imaging (EPI) sequence (TR = 2000 ms, TE = 30 ms, FOV = 240 mm, FA = 80°, 80 × 80 matrix, 3 × 3 × 3.2 mm voxels, 34 interleaved ascending axial slices, phase encoding posterior » anterior). In total, four runs of each 268 measurements were acquired. Pre-processing was carried out using the Statistical Parametric Mapping (SPM 12, https://www.fil.ion.ucl.ac.uk/spm). The first five volumes of each run were removed, the different acquisition timing of each slice was corrected and realigned to the first volume, and nonlinear distortions related to the head motion were corrected by unwarping. The high-resolution anatomical image was segmented and co-registered with the functional images to generate the skull-stripped structural image. The functional images were normalized to MNI space, interpolated to 2 × 2 × 2 mm³ voxel size, and smoothed by an 8-mm full-width at half maximum (FWHM) Gaussian kernel. Image intensity outliers were identified based on meeting any of the following criteria: (a) signal intensity >3 standard deviations from the global mean, (b) signal intensity and Mahalanobis distances >10 mean absolute deviations based on moving averages with a full-width at half maximum (FWHM) of 20 image kernels. The time points marked as outliers were severed as separate nuisance covariates included in the first-level model. Additionally, the design matrix included negative feeling ratings (1 – 9) as a parametric modulator for the reaction or regulation period.

Validation cohort. MRI data were collected on a 3.0 T scanner (Vida; Siemens Healthineers, Erlangen, Germany) with a 64-channel head coil. Structural images were acquired using high-resolution T1-weighted images by the original three-dimensional Single-shot TurboFLASH sequence from Siemens (TR = 2300 ms, TE = 2.32 ms, FOV = 240 mm, FA = 8°, 256 × 256 matrix, 0.9 × 0.9 × 0.9 mm voxels, 192 sagittal slices, phase encoding posterior » anterior) and were used for anatomical localization and warping to the standard MNI space

only. Functional images were acquired with an echo planar imaging-free induction decay (EPI-FID) sequence (TR = 2000 ms, TE = 29 ms, FOV = 240 mm, FA = 90°, 80 × 80 matrix, 3 × 3 × 3 mm voxels, 36 interleaved ascending axial slices, phase encoding posterior » anterior). In total, four runs of each 268 measurements were acquired. Using an identical pre-processing procedure to the discovery cohort (only modifying the parameters based on the sequence setting).

## Mass-univariate analysis
Discovery cohort. Subject-level general linear model (GLM) analysis was conducted using SPM12. Twenty-four head motion parameters and indicator vectors of outlier time points were modeled as non-interested regressors[54]. The fixation-cross epochs served as the implicit baseline, the periods of cue presentation and rating were modeled to eliminate the extra effects on the baseline, and the clips presented period were used as the interest regressor. A high-pass filter of 180 s was applied. Additionally, the subjective negativity ratings (1 – 9) for each clip were included as a parametric modulator for the reaction and regulation period. The primarily interesting contrasts are NR vs. NV, NA vs. NV, and NV vs. NeutV. To avoid the possibility that participants may subconsciously use the corresponding ER strategy at different runs, which may lead to confounding effects of another ER strategy when comparing NA or NR with NV, we modeled NV in the runs with NA and NR, respectively. The contrast beta images were submitted separately to the group-level analysis and used to perform MVPA analysis.

We also conducted a single-trial analysis for MVPA analysis by specifying a GLM design matrix with separate interest regressors for each clip. Nuisance regressors, implicit baseline, and high-pass filters were identical to the above analysis.

Validation cohort. The processing procedure is identical to the discovery cohort, except that the NV condition was modeled as a whole since all NV trials were in one run for the paradigm of the validation cohort.

## Multivariate pattern analysis
Following the previous framework[152], to obtain the robust brain activity patterns that can distinguish and predict between different contrasts mentioned in the mass-univariate analysis section, we trained the SVM classification (kernel linear C = 1) decoders by applying LOSO-CV methods using subject-level whole-brain contrast images data (gray matter masked[54]). Of note, only the data from the discovery cohort were used to develop the decoders. To assess the cross-validated performance of the decoders, we calculated the predictive accuracy, sensitivity, and specificity by comparing the prediction and true outcome (threshold=0). Additionally, to test the generalizability of the developed decoders, we applied them to other independent datasets, e.g., the validation cohort and generalization cohort (see 'Generalization cohort' section below), to test the pattern response for each map and the classification capacity.

## Test generalizability with independent datasets
To test our decoders' generalizability across samples, MRI systems, culture, and stimuli (dynamic versus static or heat-induced pain), we test our decoders' predictive performance among independent datasets. One recent study[34] explored the ER process and provided subject-level univariate-beta maps from the Adult Health and Behavior project–phase 2 and the Pittsburgh Imaging Project[153]. Briefly, 358 participants performed ER tasks with picture material. Reappraisal and acceptance were both reported to be used during the scanning[34]. NERS-A and NERS-R were applied to predict 'Regulate negative' vs. 'Look negative', and NNES to predict 'Look negative' vs. 'Look neutral'. Another independent picture-based ER research[65] provided data from 45 participants while using the distancing strategy (a form of reappraisal[72]) to negative pictures, as well as viewing negative and

neutral pictures. NERS-A and NERS-R were tested to classify distancing negative pictures vs. viewing negative pictures; NNES were tested to distinguish viewing negative and neutral pictures.

We also test whether developed decoders could predict using the reappraisal-like strategy to "decrease" vs. "experience" the pain induced by heat stimuli, based on the data from the generalization cohort 3 ($n$ = 33)[66]. To control the interference of visual processing caused by different materials, we excluded the voxels from the visual network[70].

**Identifying the neural basis of acceptance and reappraisal, as well as negative emotion response**
Next, we combined multiple analytic approaches to systematically interrogate the role of various brain regions or systems in acceptance and reappraisal.

Bootstrap tests and encoding model estimate. To identify robust significant features, we first bootstrapped the discovery cohort of 10,000 samples (with replacement), calculated the two-tailed uncorrected P-value of the predictive weights, and used the False Discovery Rate (FDR) correction to control multiple comparisons (set FDR $q < 0.05$). The weights of a multivariate backward/decoding model can be misleading if interpreted as reflecting brain activation of interest (e.g., related to ER). For example, a voxel that does not contain relevant information may still gain a large absolute weight because of its contribution to counteract shared noise with other voxels that do contain meaningful signals[73,154]. To address this issue and facilitate interpretation, we followed the approach proposed by Haufe and colleagues[73] and transformed the bootstrapped within-subject pattern into 'activation patterns' (forward/encoding model) using the following formula:

$$A = \text{cov}(X) \times W \times \cos(S)^{-1}$$

where $A$ is the reconstructed activation pattern, cov($X$) is the covariance matrix of training data, $W$ is the pattern weight vector, and cov($S$) is the covariance matrix of the latent factors, which is defined as $W^T \times X$. This reconstructed activation obtained via the Haufe transformation is analogous to 'structure coefficients', providing interpretable predictive-feature values that reflect the direction of the association between voxel activity and the target variable, indicating whether higher activation predicts greater or lower engagement in the corresponding mental process (e.g., the use of emotion regulation strategies)[54,73,155]. It has been shown to improve both the interpretability and reliability of the predictive features[155–158]. The group-level significant brain regions were obtained by conducting a one-sample $t$-test (FDR $q < 0.05$).

To provide a more comprehensive understanding of the relationship between the reconstructed activation patterns and the decoding weight maps, we also analyzed the voxel-wise spatial similarity between their normalized unthresholded maps for each strategy separately.

Bayes Factor analysis. To identify the brain regions that are acceptance or reappraisal specific or the common regions of those two strategies, we utilized the BF approach to quantify evidence for alternative and null hypotheses[34,67]. Specifically, the BF value of a given voxel reflects the likelihood of the alternative and null (e.g., the neural activity is correlated with naturalistic acceptance or not). The BF values are calculated from $t$-values of a one-sample $t$-test of encoding maps from the discovery cohort, using a Jeffreys-Zellner-Siow (JZS) prior and formula (see below) provided by a previous study[159].

$$B_{01} = \frac{\left(1 + \frac{t^2}{v}\right)^{-(v+1)/2}}{\int_0^\infty \left(1 + Ngr^2\right)^{-\frac{1}{2}}\left(1 + \frac{t^2}{(1+Ngr^2)v}\right)^{-\frac{v+1}{2}}(2\pi)^{-1/2}g^{-3/2}e^{-1/(2g)}dg}$$

where $N$ represents the sample size, $t$ for the $t$-statistic, $v$ denotes the degrees of freedom, and $r$ is the scale factor. In the current study, r was set as 0.707, which suggested a moderate effect size[34,159].

Because the null is bound by t = 0 and the current sample size (N = 59), the BF cannot be smaller than 0.14 but can be infinitely large; a heuristic threshold of 5 was set for the alternative hypothesis and 1/5 for the null to attain moderate to strong evidence in favor of both hypotheses[67,160]. Note that in current datasets, the $P$ value threshold of BF = 5 is $P$ = 0.0067, which is smaller than the $q < 0.05$ FDR-corrected threshold for the encoding model of NERS-R ($P$ = 0.0190) but not for NERS-A ($P$ = 0.0051), hence we further bound the alternative results by the corresponding FDR-corrected threshold, specifically as follows:

- 'Acceptance only' voxels—positive activation with BF > 5 and $P$ < 0.0051 for acceptance, and BF < 1/5 for reappraisal.
- 'Common' voxels—positive activation with BF > 5 and $P$ < 0.0051 for acceptance, and with BF > 5 and $P$ < 0.0190 for reappraisal.
- 'Reappraisal only' voxels—positive activation with BF > 5 and $P$ < 0.0051 for reappraisal, and BF < 1/5 for acceptance.

To verify the results of BF analysis, we also performed the conjunction analysis between the thresholded (FDR q < 0.05) activation maps of those two strategies obtained from mass-univariate analysis and the reconstructed "activation pattern", separately.

Spatial similarity between stable decoding maps and interest networks, as well as crucial ROIs. The spatial similarity between stable encoding maps and seven large-scale resting-state networks[70], as well as crucial prefrontal ROIs[69] (dlPFC, vmPFC, dmPFC, vlPFC, and ACC), was illustrated by river plots. The spatial similarity was computed as cosine similarity between the networks or ROIs and the threshold NERS-A, NERS-R, and NNES (FDR $q < 0.05$, positive values only), reflecting the concordance between each model and corresponding brain responses[161]. To further compare the underlying neural representations between NERS-A and NERS-R, we examined the voxel-level spatial covariation between their unthresholded reconstructed activation patterns and weight maps for those two models (z-scored), separately.

Comparing the performance of developed models with PINES. To investigate whether developed decoders capture both the common and separate aspects of naturalistic acceptance, reappraisal, and reaction to negative emotion, we compared the decoders' performances on data of different conditions from the validation cohort and along with a decoder provided by a previous study[81] – PINES – which has been demonstrated could track general negative emotion experiences induced by pictures.

**Evaluation of predictive performances of local systems**
To investigate the contribution of a single resting-state brain network (e.g., DMN) or individual regions to the ER or react processing, we first estimated the prediction performance of seven large-scale networks as well as the prefrontal cortex, which has been demonstrated to have crucial roles in both generating and regulating emotions[32,33,36] and compared the performance with developed models. Additionally, we employed a whole-brain searchlight-based analysis (three-voxel radius spheres) to test the predictive performance of individual regions.

**Clinical application potential**
ER and reaction abnormalities have been demonstrated in MDs, e.g., SUDs[10,71,85,131], and may serve as a transdiagnostic structure[5,7–11]. To examine whether our decoders could detect the emotion-related abnormality among SUD individuals, we applied NERS-A and NERS-R to classify the distancing (one of reappraisal) strategies with negative emotions induced by negative pictures, and also applied NNES to predict negative vs. neutral emotions. The data were provided by a previous study[71] and newly collected data using a similar ER task paradigm and the identical scanner and sequence settings (details see

ref.[71] and supplementary, study protocol NCT02801214, clinicaltrials.gov). Briefly, 48 male health control participants (HC, 18 from our previous study[71] and 30 newly collected, 24.3 ± 5.0 years old) and 49 male cannabis users (CU, 23 from our previous study and 26 newly collected, 24.1 ± 5.3 years old) were included in the final analysis. All participants provided written informed consent prior to participation. Both previously published and newly collected data were obtained under the same ethical approval granted by the Medical Faculty of the University of Bonn and conducted in accordance with the Declaration of Helsinki. We further used the permutation test to investigate whether the pattern expression of the decoders to the data of HC and CU has a significant group difference[142,143]. In this approach, the $H_0$ assumes that the HC and CU groups have no significant differences. In each permutation, individual-level data were randomly sampled from the two groups, standardized using Z-score normalization to mitigate the influence of potential outliers, and subsequently subjected to forced-choice classification, with a total of 10,000 permutations performed. Finally, we tested whether the observed classification accuracy of the original group (e.g., HC) differed significantly ($p < 0.05$) from the null distribution generated by permutation.

### Reporting summary
Further information on research design is available in the Nature Portfolio Reporting Summary linked to this article.

## Data availability
The fMRI data generated in this study have been deposited in the figshare repository under: https://doi.org/10.6084/m9.figshare.29179934. The fMRI data used in the generalization cohorts are available in the NeuroVault (https://neurovault.org/collections/16266)[34], OpARA (https://opara.zih.tu-dresden.de/xmlui/handle/123456789/1951)[65], and OpenfMRI (https://openfmri.org/dataset/ds000140)[66]. Source data are provided with this paper.

## Code availability
Code for analyzing data and generating figures is available via GitHub at https://github.com/canlab and https://github.com/h-psy/fMRI_studies/tree/main/NERS. A version of the code used in this study is available via Zenodo (https://doi.org/10.5281/zenodo.18625297)[162].

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

## Acknowledgements

This work was funded by the Brain Science and Brain-like Intelligence Technology-National Science and Technology Major Project (No.2022ZD0208500, to DY) and Sichuan Science and Technology Program (2024ZDZX0014, to DY) (National Natural Science Foundation of China (Grant No. 82271583, to B.B.; No. 32300862, to F.Z.), start-up/seed grant from the University of Hong Kong to B.B., the Chongqing social science planning project (2024YC035) to F.Z., the Natural Science Foundation of Chongqing (CSTB2023NSCQ-MSX0889) to F.Z., and was supported by the China Scholarship Council Program (No. 202506070094, to H.J.), the Sichuan Science and Technology Program (2023YFS0023, to B.Z.; 24NSFSC6564, to L.L.). Funders were not involved in the research design, the collection and analysis of data, the decision to publish, or the writing of the manuscript. We thank the authors of Bo, K. et al., Woo, C.-W. et al., and Diers, K. et al. for providing data for the generalization cohorts in the present study, and we thank the authors of Chang, L. J. et al. for providing the PINES signature.

## Author contributions

H.J. and B.B. were responsible for the research design and drafting of the manuscript. H.J., J.H., K.Z., B.Z., L.L., S.F., L.W., and X.Z. collected the data. H.J., F.Z., and B.B. were responsible for the analysis and interpretation of data. X.G., K.M.K., W.Z., D.Y., and T.Y. provided important suggestions and tools for the analyses and critically commented on the manuscript. B.B. supervised the project and acquired the funding. All authors meet the ICMJE's four criteria for authorship and are responsible for revising the paper, approving the final version for publication, and ensuring the accuracy and completeness of the work.

## Competing interests

The authors declare no competing interests.

## Additional information

¹Sichuan Provincial Center for Mental Health, Sichuan Provincial People's Hospital, School of Medicine, University of Electronic Science and Technology of China, Chengdu, China. ²Ministry of Education Key Laboratory for Neuroinformation, School of Life Science and Technology, University of Electronic Science and Technology, Chengdu, China. ³Independent Researcher, Cologne, Germany. ⁴Institute of Brain and Psychological Sciences, Sichuan Normal University, Chengdu, China. ⁵Shanghai Key Laboratory of Psychotic Disorders, Brain Health Institute, National Center for Mental Disorders, Shanghai Mental Health Center, Shanghai Jiaotong University School of Medicine and School of Psychology, Shanghai, China. ⁶Co-innovation Center of Neuroregeneration, Nantong University, Nantong, Jiangsu, China. ⁷Faculty of Psychology, Southwest University, Chongqing, China. ⁸MOE Key Laboratory of Cognition and Personality, Chongqing, China. ⁹MIND & AI Lab, Department of Psychology, The University of Hong Kong, Hong Kong, China. ¹⁰SRT AI, Society & Social Dynamics, Faculty of Social Sciences, The University of Hong Kong, Hong Kong, China. ✉e-mail: zhou.feng@live.com; bbecker@hku.hk

