## [Transparent Peer Review file · Nature Communications]

Common and distinct neurofunctional signatures of dynamic naturalistic emotion regulation strategies

Corresponding Author: Professor Benjamin Becker

Version 0:

Reviewer comments:

Reviewer #1

(Remarks to the Author)

The authors have satisfactorily addressed many of the comments and suggestions I raised regarding the initial version of the manuscript. Thanks for that.

First, as suggested, the authors compared the decoding model weights with the reconstructed activation patterns, as shown in Extended Data Fig. 1. While this analysis is informative, it would be helpful if the authors could further indicate which specific brain voxels drive the observed positive correlations—i.e., those corresponding to regions 1 and 6 in the scatter plots.

Second, the authors applied the Haufe transformation across the training, validation, and test datasets, with the results presented in Extended Data Fig. 2. This addition strengthens the study, but there is only a minimal description about the new results. A more detailed description and interpretation of the comparisons across these datasets would improve clarity and help readers better understand the implications of these results.

Reviewer #2

(Remarks to the Author)

The authors have addressed all my comments very well and I have no further concerns. I think this is excellent and impressive amount of work that should be published.

Reviewer #3

(Remarks to the Author)

The authors have revised and improved aspects of the paper in response to previous comments on the originally submitted version. These changes have clarified some previous issues and filled in some important missing information. However, the responses to some important issues that were raised do not appear to fully engage with the concerns and in one case no clear response was provided. My overall sense of the study is that it addresses a very worthwhile issue and makes some headway on each of the four issues the authors note as remaining unresolved (see line 74), but when these component advances are combined, the overall level of novelty and theoretical and applied significance falls short of what one would expect from the journal, and therefore the paper seems more appropriate for a more specialist journal.

1. As an example of the incremental nature of the advances in the paper, the prior review comment stated: “For example, it is perhaps not surprising that two strategies that differ greatly in the cognitive processes they involve, one involving effortful, active, top-down regulation of emotion and the other involving passive meta-cognitive awareness and self-referential processes, have both overlapping and non-overlapping neural correlates.”. I couldn’t find a specific response to this concern, other than to reiterate that the strategies are “fundamentally contrasting emotion regulation strategies”. The authors’ response focuses on the limitations of other studies and approaches and the advantages of the application of multivariate machine-learning methods in the current study. It’s left unclear whether the authors agree with the comment’s implication that this concern, if valid, suggests that this major aspect of the study is relatively incremental.

2. Regarding the point about the novelty and importance of using naturalistic stimuli, the authors acknowledge that they don’t

have data directly showing this. Absent this, they point to the relatively low accuracy of PINES and evidence that picture-based fMRI signatures have shown limited generalizability to dynamic emotional stimuli. However, there are multiple reasons why PINES may exhibit lower prediction accuracy other than the static vs. naturalistic dimension (e.g., differences in the analytic pipeline and approach) so this claim (the basis of one of the 4 key unresolved issues noted by the authors as motivating the study) regarding the key importance of using naturalistic stimuli remains speculative rather than directly supported by evidence.

3. The response to the point about the incremental nature of the clinical translational potential of the findings includes some additional new analyses, which is a welcome addition. Out of all their responses, I think this is most effective response from the authors. However, in the context of the current literature, much of which the authors cite in their response, even with these additional findings, it's still not clear that crosses the line to being more than an incremental advance. The newly revised discussion (around line 627) also contributes to the sense that this advance is only one step on the road, with much more work required for validation and generalization.

4. For their response to the comment about the challenges top-down models (e.g. Zhang), the authors point out that the paper had not been published when their paper had been submitted, but challenges to the currently dominant top-down models of emotion regulation have been around for several years, with some of these papers being referenced in that review article (see for example references 17-19 in Zhang et al.) – thus, the antecedent theoretical views that were elaborated in Zhang et al. have been available in the literature for years and should be addressed to an appropriate extent. The authors now briefly acknowledge this other major theoretical perspective, but don't go on to discuss these other perspectives and implications for the interpretation of their findings (though the review comment requested discussion of the implications). This issue is relevant to the overall goal of not only providing predictive models, but just as importantly, explaining the overall findings in the context of current theoretical models. If one adopts the alternative perspective described by Zhang et al., the interpretation of the current findings would be quite different. Ideally, additional analyses could help establish the extent to which the findings conclusively support the authors' theoretical claims.

Dear Reviewers:

We thank the reviewers for their positive evaluation of our revision and the additional comments that allowed us to further improve our manuscript. The comments allowed us to improve the clarity, rigor, and overall quality of the manuscript.

For your reference, we provide two versions of the manuscript: a **clean version** incorporating the reviewers' comments, and a **tracked-changes version** highlighting all the revisions made in this round. Detailed responses to each point are provided below.

With sincere regards

Reviewers' Comments and Authors' Responses:

Reviewer #1 (Remarks to the Author):

The authors have satisfactorily addressed many of the comments and suggestions I raised regarding the initial version of the manuscript. Thanks for that.

Response:

We thank the reviewer for the positive evaluation of the revised manuscript. We appreciate the reviewer's constructive feedback and comments on the original version, which has greatly helped to improve the manuscript.

First, as suggested, the authors compared the decoding model weights with the reconstructed activation patterns, as shown in Extended Data Fig. 1. While this analysis is informative, it would be helpful if the authors could further indicate which specific brain voxels drive the observed positive correlations—i.e., those corresponding to regions 1 and 6 in the scatter plots.

Response: We thank the reviewer for this follow-up suggestion and fully agree that further specifying the voxels that drive the positive association will be important to set the findings better into context. Based on the suggestion, we have revised Extended Data Fig. 1, which now explicitly shows which specific brain voxels drive

the observed positive correlations between the encoding and decoding models. In the updated figure, the regions highlighted in the scatter plots are now displayed on the standard reference brain; for completeness, we have shown both the shared positive voxels (Octants 2, shown in yellow) and the shared negative voxels (Octants 6, shown in blue). The resulting spatial patterns largely overlap with those obtained by the FDR-corrected Haufe transformation and, as such, further confirm our findings. We have included the new information and interpretation as follows:

The related context after revision is as follows:

Extended Data:

(Line 923 in the tracked-changes version; the same applies below)

Extended Data Fig.1 | Voxel-level spatial similarity between decoding and encoding models. Voxel-level spatial similarity between the normalized unthresholded decoding weighted map and encoding map of NERS-A (a), NERS-R (b), and NNES (c), separately. Scatter plots illustrate normalized unthresholded voxel beta weights of the decoding map (y-axis) and encoding map (x-axis). Colored octants indicate voxels of shared positive (Octants 2) or shared negative (Octants 6), selective positive weights for decoding map (Octant 1) or encoding map (Octant 3), selective negative weights for decoding map (Octant 5) or encoding map (Octant 7), and voxel weights opposite for the two maps (Octants 4 and 8). The middle bars indicate the sum of squared distances

from the origin (0, 0) for each octant, which integrates the number of voxels and combined weights. Displayed on the right are the voxel distributions of the patterns with consistent positive correlations, i.e., voxels originating from the shared positive (Octants 2, shown in yellow) and shared negative (Octants 6, shown in blue) sectors. The resulting spatial patterns are largely consistent with and cover a broader spatial extent than those identified by the FDR-corrected Haufe transformation.

Second, the authors applied the Haufe transformation across the training, validation, and test datasets, with the results presented in Extended Data Fig. 2. This addition strengthens the study, but there is only a minimal description about the new results. A more detailed description and interpretation of the comparisons across these datasets would improve clarity and help readers better understand the implications of these results.

Response:

We thank the reviewer for the constructive suggestion and agree that the results will benefit from a better integration into the manuscript. We have therefore added more detailed descriptions and interpretations, along with a brief discussion of the corresponding results in the main text, to improve clarity and integration of the findings.

The related context after revision is as follows:

Results section:

(Line 306)The effects were confirmed by the reconstructed activation patterns across datasets and with univariate analyses (Extended Data Figs. 2 and 3). Consistent reconstructed patterns were observed under conditions where the decoders achieved significant predictions (e.g., NERS-R across all datasets), demonstrating the robustness of our findings. For the NERS-A, the generalization to cohorts 2 and 3 remained inconsistent, likely reflecting that the Haufe-transformed patterns were derived from data based on a different emotion regulation process than that used to train the respective acceptance-tracking decoder (i.e., cohorts 2 and 3 employed emotion regulation via reappraisal rather than via acceptance – the strategy used in the NERS-A data). Together, these observations further support the robustness and, to a certain extent, process-specificity of the developed decoders.

Extended Data:

(Line 941)

Extended Data Fig.2 | Reconstructed 'activation pattern' from the validation and generalization cohorts. FDR corrected ($q < 0.05$) group-level reconstructed 'activation pattern' transformed from the validation cohort (**a**), generalization cohort 1 (**b**), generalization cohort 2 (**c**), and generalization cohort 3 (**d**). The color indicates the direction of the relationship between each voxel and the target variable – i.e., which voxels are positively (red) or negatively (blue) related to the corresponding mental processing (e.g., emotion regulation strategies). Consistent reconstructed activation patterns were observed in datasets where the decoders achieved significantly accurate predictions (e.g., NERS-R across all datasets), indicating the robustness of the findings. Some inconsistencies were noted, for example, in generalization cohorts 2 and 3 for

NERS-A, possibly because the reconstructed ‘activation patterns’ were derived from data corresponding to a different mental process than that used to train the decoder (i.e., reappraisal rather than acceptance).

Discussion section:

(Line 489) The robustness of these observations is further supported by the consistent reconstructed brain activation patterns across multiple datasets and traditional univariate analyses. Of note, the NERS-A, trained on experimental data using the acceptance strategy, did not generalize to experimental data obtained during emotion regulation via reappraisal (generalization cohorts 2 and 3), which underscores – to a certain extent – the process-specificity of the neural decoders.

Reviewer #2 (Remarks to the Author):

The authors have addressed all my comments very well and I have no further concerns. I think this is excellent and impressive amount of work that should be published.

Response:

We sincerely thank the reviewer for the positive evaluation and encouraging comments. We appreciate the time, effort, and thoughtful evaluation of our work.

Reviewer #3 (Remarks to the Author):

The authors have revised and improved aspects of the paper in response to previous comments on the originally submitted version. These changes have clarified some previous issues and filled in some important missing information. However, the responses to some important issues that were raised do not appear to fully engage with the concerns and in one case no clear response was provided. My overall sense of the study is that it addresses a very worthwhile issue and makes some headway on each of the four issues the authors note as remaining unresolved (see line 74), but when these component advances are combined, the overall level of novelty and theoretical and applied significance falls short of what one would expect from the journal, and therefore the paper seems more appropriate for a more specialist journal.

Response:

We thank the reviewer for acknowledging the improvements made to the manuscript and for noting that the revisions have helped to clarify some previously unclear points and to address important missing information. We also apologize that some of our earlier responses in revision 1 did not clearly indicate the corresponding changes in the manuscript, and we have included further revisions based on the new comments. For details, please refer to the detailed responses below, with clear indications of the corresponding revisions in the manuscript.

Moreover, we appreciate the reviewer's recognition that the study addresses an important open question and makes progress toward the unresolved issues outlined in our revised Introduction. With respect to the concern regarding overall novelty and theoretical and applied significance, we would like to clarify that from our perspective, the manuscript's contribution lies in jointly addressing multiple unresolved questions within a single, coherent experimental and analytical framework. Specifically, the work integrates several open issues in the literature through a unified line of investigation that combines advanced multivariate analytic approaches with new experimental data from multiple independent samples, including healthy individuals (a preregistered discovery cohort, $n = 59$; and independent preregistered validation cohort, $n = 33$; other independent generalization cohorts from other labs, $n = 358, 45, \text{ and } 33$, respectively) as well as extended data from cannabis users ($n = 49$; healthy controls, $n = 48$). We believe that this combination of methodological rigor, cross-sample validation, and the resulting conceptual and applied implications of the findings, provides a level of contribution that is appropriate for the scope and readership of *Nature Communications*.

1. As an example of the incremental nature of the advances in the paper, the prior review comment stated: "For example, it is perhaps not surprising that two strategies that differ greatly in the cognitive processes they involve, one involving effortful, active, top-down regulation of emotion and the other involving passive meta-cognitive awareness and self-referential processes, have both overlapping and non-overlapping neural correlates.". I couldn't find a specific response to this concern, other than to reiterate that the strategies are "fundamentally contrasting emotion regulation strategies". The authors' response focuses on the limitations of other studies and approaches and the advantages of the application of multivariate machine-learning

methods in the current study. It's left unclear whether the authors agree with the comment's implication that this concern, if valid, suggests that this major aspect of the study is relatively incremental.

Response:

We thank the reviewer for raising this point and apologize that our previous response did not clearly indicate how we addressed this specific example in the previous revision. While we agree that, at a general level, it may not be surprising that two emotion regulation strategies with markedly different cognitive operations show both overlapping and non-overlapping neural correlates, we believe that a theoretical assumption alone is not sufficient. Crucially, as highlighted by the other reviewers, the substantial differences in the operational and cognitive mechanisms of reappraisal and acceptance make them particularly well-suited for a direct experimental comparison within the same sample. Such a comparison, within a pre-registered, advanced analytical and replication–generalization design, is important for advancing our understanding of emotion regulation processes beyond intuition or post hoc inference.

Extensive revisions in the second paragraph of the Introduction aimed at addressing this point raised during the previous revision from a conceptual and theoretical perspective by clarifying why a direct, within-subject comparison of reappraisal and acceptance is informative and critical for determining competing accounts^{1,2}. In addition, in response to Comment 4, we have now further expanded this section to explicitly acknowledge recent challenges to traditional top–down and strong separability models (e.g., Zhang et al.).

In summary, the contribution of the present work is not to demonstrate that overlap and distinction between emotion regulation strategies coexist, but to empirically clarify the representational level at which strategy differences emerge under dynamic and naturalistic contexts —an issue that cannot be resolved by theoretical intuition alone (or by univariate analyses of fMRI data or the corresponding meta-analyses based on univariate results, see also for methodological discussions^{3,4}). Together with existing empirical gaps and methodological constraints in the literature, these aspects position the current study as addressing an important and timely open question. Against this background, we respectfully disagree that our work should be considered merely “relatively incremental.” We have revised corresponding parts in the manuscript to outline the novelty and relevance of our work as follows:

Introduction section:

(Line 34) While both strategies focus on cognitive changes⁵ (see *process model of emotion regulation*^{6,7}), they differ markedly in their operational and cognitive mechanisms. Reappraisal is based on reinterpreting the subjective meaning of the emotion-inducing event to control its affective impact^{8,9}. It critically relies on multiple executive functions, including attention, working memory, and cognitive control, which represent an active and cognitively demanding process^{1,10}. While reappraisal is the most extensively studied ER strategy, a critical question remains: what constitutes an effective and psychologically distinct alternative? Strategies like avoidance, suppression, or distraction have been commonly investigated; however, they are often maladaptive and associated with poor long-term outcomes^{11,12}. In contrast, acceptance, a core component of mindfulness-based third-wave cognitive therapies, involves nonjudgmental awareness of emotional experiences without attempts to control them^{13,14}. It has been recognized as both a potent and adaptive alternative, and is considered a less cognitively demanding and more passive strategy relying on self-referential processes and metacognitive awareness¹. This distinction is crucial for both cognitive neuroscience and therapeutic applications. If reappraisal and acceptance simply recruit the same neural pathways, it would suggest a final common pathway for effective regulation. Conversely, dissociable neural patterns would provide strong evidence for multiple, distinct routes to ER and adaptive mental health, with significant implications for personalized interventions. Additionally, ongoing debates¹⁵⁻¹⁷ about the extent of neural overlap between emotion generation and regulation (as well as the overlap between different emotion regulation strategies¹⁸) further highlight the need for empirical work that initially carefully characterizes the neural basis of distinct emotion regulation processes. Directly comparing reappraisal and acceptance represents as such a critical step for characterizing strategy-specific neural substrates involved in the regulation of negative affect.

2. Regarding the point about the novelty and importance of using naturalistic stimuli, the authors acknowledge that they don't have data directly showing this. Absent this, they point to the relatively low accuracy of PINES and evidence that picture-based fMRI signatures have shown limited generalizability to dynamic emotional stimuli. However, there are multiple reasons why PINES may exhibit lower prediction accuracy

other than the static vs. naturalistic dimension (e.g., differences in the analytic pipeline and approach) so this claim (the basis of one of the 4 key unresolved issues noted by the authors as motivating the study) regarding the key importance of using naturalistic stimuli remains speculative rather than directly supported by evidence.

Response:

We thank the reviewer for this comment. As noted in the manuscript (Line 123), numerous recent reviews, perspectives, and empirical fMRI studies support that naturalistic paradigms, with their dynamic nature, better approximate real-life experiences, including affective experiences, and can increase the ecological validity of neural models^{19–26}. Of note, understanding how brain systems supporting emotion generation and regulation operate in everyday life, in naturalistic settings, is highlighted as one of the outstanding questions in the recent Opinion paper mentioned by this reviewer (see Zhang et al.).

However, on a methodological level, we also agree that other factors may influence the performance of decoders trained on static images (PINES), and we have therefore included a carefully qualified statement in the Discussion (Line 521) to avoid overinterpretation of these findings. Nevertheless, our results demonstrate clear performance differences between decoders trained on static images (PINES) and those trained under naturalistic paradigms in the present study (NNES). Along with the results from the current study, findings of related studies from our team indicate that signatures derived from dynamic stimuli generalize well to data based on static stimuli²⁷, whereas signatures derived from data based on static stimuli show poor generalization to dynamic contexts²⁴. While previous work²⁰ has compared classical versus naturalistic cognitive neuroscience paradigms and highlighted the advantages of naturalistic approaches; a direct comparison of emotion regulation processes under dynamic versus static stimuli is beyond the primary focus of the present study. However, we also agree that a detailed and ultimate comparison would require the acquisition of novel data with closely aligned experimental instructions, acquisition parameters, etc. To provide the reader with a more balanced discussion of the findings and clearly underscore the limitations regarding this point we have revised the corresponding discussion sections as follows:

The related context after revision is as follows:

Discussion section:

(Line 517) The relatively poor performance of PINES when predicting data based on naturalistic dynamic stimuli may also underscore that the neural systems for processing emotional information in dynamic context may differ from those engaged by sparsely presented stimuli in conventional laboratory experiments (see ref.^{15,23,24}). Although our results indicate differences between static and naturalistic paradigms, we do not claim that naturalistic stimuli are inherently superior, as multiple factors may contribute to these observed differences.

(Line 642) While previous studies have demonstrated the advantages of naturalistic paradigms in terms of ecological validity^{19–26}, further research directly comparing reappraisal and acceptance processes under naturalistic paradigms versus static-image-based paradigms are required to directly compare the neural processes underlying different emotion regulation processes in response to static (e.g., pictures) and dynamic (e.g., movies) stimuli.

3. The response to the point about the incremental nature of the clinical translational potential of the findings includes some additional new analyses, which is a welcome addition. Out of all their responses, I think this is most effective response from the authors. However, in the context of the current literature, much of which the authors cite in their response, even with these additional findings, it's still not clear that crosses the line to being more than an incremental advance. The newly revised discussion (around line 627) also contributes to the sense that this advance is only one step on the road, with much more work required for validation and generalization.

Response:

We thank the reviewer for the positive evaluation of the additional analyses addressing the clinical translational potential of the findings. While further validation and generalization are indeed necessary to fully translate neuroimaging into application, this does not imply limited relevance of the present work.

Altered emotional reactivity and deficits in emotion regulation are well-established, transdiagnostic features across mental disorders, providing a strong conceptual basis for identifying neural markers of dysfunctional emotion regulation processes^{12,28–32}. One recent comprehensive review³³ highlights that machine-learning-based approaches can yield mechanistically informative neuromarkers with clear

clinical relevance, a view supported by initial and innovative high-impact studies in addiction³⁴ and depression³⁵.

Importantly, to our knowledge, the present study is the first to test the clinical applicability of a multivariate neuromarker for emotion regulation in a clinical population. The approach is based on a recent innovative study, which aimed at translating an fMRI MVPA-based neuromarkers for drug-cue reactivity into clinical application³⁴. The work and a flanking commentary³⁶ – both published in *Nature Neuroscience* (include the commentary) – emphasize both the promise of distributed neural signatures as transdiagnostic biomarkers and the need for continued validation, an emphasis that reflects standard translational practice rather than a limitation unique to the present study.

Together, the innovative, comparative, and strategy-specific approach in the current study extends the literature beyond incremental advances focused on single strategies, and represents a substantive step toward clinically actionable and process-specific neuromarkers of emotion regulation.

4. For their response to the comment about the challenges top-down models (e.g. Zhang), the authors point out that the paper had not been published when their paper had been submitted, but challenges to the currently dominant top-down models of emotion regulation have been around for several years, with some of these papers being referenced in that review article (see for example references 17-19 in Zhang et al.) – thus, the antecedent theoretical views that were elaborated in Zhang et al. have been available in the literature for years and should be addressed to an appropriate extent. The authors now briefly acknowledge this other major theoretical perspective, but don't go on to discuss these other perspectives and implications for the interpretation of their findings (though the review comment requested discussion of the implications). This issue is relevant to the overall goal of not only providing predictive models, but just as importantly, explaining the overall findings in the context of current theoretical models. If one adopts the alternative perspective described by Zhang et al., the interpretation of the current findings would be quite different. Ideally, additional analyses could help establish the extent to which the findings conclusively support the authors' theoretical claims.

Response:

We thank the reviewer for emphasizing the importance of the alternative perspectives that challenge top-down models, including Zhang et al. and related earlier work (references 17–19). We acknowledge that these perspectives warrant further consideration and have expanded the *Introduction* and *Discussion* to situate our findings within this broader framework.

Viewed through this lens, our results remain interpretable and informative. For example, shared contributions of the default mode network across strategies align with common valuation- and meaning-related processes, while strategy-specific engagement of frontoparietal control networks during reappraisal and amygdala–somatomotor–attention systems during acceptance reflects preferential recruitment of distinct subprocesses within a shared architecture. Thus, our findings do not rely on a strict neural separability assumption, but provide empirical support for shared-and-distinct subprocess accounts (and in line with the conclusion of the Zhang et al paper regarding one of the two key datasets the papers is based on – Bo et al., - suggesting that: “It is worth noting that while most brain regions show overlap between emotion regulation and emotion generation, this study [77] also identified a set of regions whose responses were consistent with the neural separability assumption, at least in the context of downregulation of responses to emotional images”).

We emphasize that the primary aim of the current study was to compare the neural representations of reappraisal and acceptance under dynamic, naturalistic conditions, rather than to test overarching theories on emotion generation versus separation. Further dissociating shared versus distinct subprocesses of emotion generation and regulation is moreover not in line with our pre-registered analyses or the design of the study and would as such require additional exploratory post hoc analyses on experimental data not designed to address this question. As such, an extensive discussion is beyond the current scope of our work. Nevertheless, by leveraging brain-wide multivariate modeling and task contrasts, our study provides a robust, theory-agnostic characterization of common and strategy-specific neural processes in the domain of emotion regulation.

Based on the comment from the reviewer, we have further expanded the corresponding discussion – in addition to changes in the introduction that have been included in the last revision to acknowledge these accounts and the recent opinion paper - to clarify how the findings can be interpreted across frameworks while maintaining

their core empirical contributions. Changes in the related context are outlined below:

The related context after revision is as follows:

Introduction section:

(Line 51) Additionally, ongoing debates^{15–17} about the extent of neural overlap between emotion generation and regulation (as well as the overlap between different emotion regulation strategies¹⁸) further highlight the need for empirical work that initially carefully characterizes the neural basis of distinct emotion regulation processes.

Discussion section:

(Line 596) Of note, our findings do not rely on a strict top–down or strong neural separability account of emotion regulation. Recent theoretical perspectives have emphasized substantial neural overlap between emotion generation and emotion regulation, proposing that both processes draw on shared valuation-related systems while preferentially engaging distinct subprocesses depending on contextual demands and regulatory goals^{15,37,38}. Interpreted within this perspective, the present results suggest that in the context of emotion regulation, acceptance and reappraisal recruit common, distributed neural systems, in particular core regions of the DMN, while also recruiting distinguishable engagement of frontoparietal, somatomotor, attentional, and subcortical systems. The observed distinguishability may, to a certain extent, reflect differences in how shared neural resources are coordinated and weighted, rather than the engagement of entirely separate neural modules.

References

(Note: Reference numbers correspond to citations in this Author Response.)

1. Goldin, P. R., Moodie, C. A. & Gross, J. J. Acceptance versus reappraisal: Behavioral, autonomic, and neural effects. *Cogn. Affect. Behav. Neurosci.* **19**, 927–944 (2019).
2. Monachesi, B., Grecucci, A., Ahmadi Ghomroudi, P. & Messina, I. Comparing reappraisal and acceptance strategies to understand the neural architecture of emotion regulation: a meta-analytic approach. *Front. Psychol.* **14**, 1187092 (2023).
3. Woo, C.-W., Chang, L. J., Lindquist, M. A. & Wager, T. D. Building better biomarkers: brain models in translational neuroimaging. *Nat. Neurosci.* **20**, 365–377 (2017).
4. Peelen, M. V. & Downing, P. E. Testing cognitive theories with multivariate pattern analysis of neuroimaging data. *Nat. Hum. Behav.* **7**, 1430–1441 (2023).
5. Willroth, E. C. & John, O. P. Assessing individual differences in emotion regulation: Habitual strategy use and beyond. in *Handbook of emotion regulation, 3rd ed* 22–30 (The Guilford Press, New York, NY, US, 2024).
6. Gross, J. J. The emerging field of emotion regulation: An integrative review. *Rev. Gen. Psychol.* **2**, 271–299 (1998).
7. Gross, J. J. Emotion regulation: Conceptual and empirical foundations. *Handb. Emot. Regul.* **2**, 3–20 (2014).
8. Buhle, J. T. *et al.* Cognitive reappraisal of emotion: A meta-analysis of human neuroimaging studies. *Cereb. Cortex* **24**, 2981–2990 (2014).
9. Gross, J. J. Antecedent- and response-focused emotion regulation: Divergent consequences for experience, expression, and physiology. *J. Pers. Soc. Psychol.* **74**, 224–237 (1998).
10. Troy, A. S., Shallcross, A. J., Brunner, A., Friedman, R. & Jones, M. C. Cognitive reappraisal and acceptance: Effects on emotion, physiology, and perceived cognitive costs. *Emotion* **18**, 58–74 (2018).
11. Troy, A. S. *et al.* Psychological Resilience: An Affect-Regulation Framework. *Annu. Rev. Psychol.* **74**, 547–576 (2023).
12. Aldao, A., Nolen-Hoeksema, S. & Schweizer, S. Emotion-regulation strategies across psychopathology: A meta-analytic review. *Clin. Psychol. Rev.* **30**, 217–237 (2010).
13. Bishop, S. R. *et al.* Mindfulness: A proposed operational definition. *Clin. Psychol.*

- Sci. Pract.* **11**, 230 (2004).
14. Kober, H., Buhle, J., Weber, J., Ochsner, K. N. & Wager, T. D. Let it be: mindful acceptance down-regulates pain and negative emotion. *Soc. Cogn. Affect. Neurosci.* **14**, 1147–1158 (2019).
 15. Zhang, J.-X., Bo, K., Wager, T. D. & Gross, J. J. The brain bases of emotion generation and emotion regulation. *Trends Cogn. Sci.* (2025) doi:10.1016/j.tics.2025.04.013.
 16. Campos, J. J., Frankel, C. B. & Camras, L. On the Nature of Emotion Regulation. *Child Dev.* **75**, 377–394 (2004).
 17. Pessoa, L. On the relationship between emotion and cognition. *Nat. Rev. Neurosci.* **9**, 148–158 (2008).
 18. Morawetz, C., Bode, S., Derntl, B. & Heekeren, H. R. The effect of strategies, goals and stimulus material on the neural mechanisms of emotion regulation: A meta-analysis of fMRI studies. *Neurosci. Biobehav. Rev.* **72**, 111–128 (2017).
 19. Saarimäki, H. Naturalistic Stimuli in Affective Neuroimaging: A Review. *Front. Hum. Neurosci.* **15**, 675068 (2021).
 20. Sonkusare, S., Breakspear, M. & Guo, C. Naturalistic Stimuli in Neuroscience: Critically Acclaimed. *Trends Cogn. Sci.* **23**, 699–714 (2019).
 21. Saarimäki, H. *et al.* Discrete Neural Signatures of Basic Emotions. *Cereb. Cortex* **26**, 2563–2573 (2016).
 22. Nanni-Zepeda, M. *et al.* Neural signatures of shared subjective affective engagement and disengagement during movie viewing. *Hum. Brain Mapp.* **45**, e26622 (2024).
 23. Zhou, F. & Becker, B. Understanding human brain function in real-world environments. *PLOS Biol.* **23**, e3003210 (2025).
 24. Zhou, F. *et al.* Capturing Dynamic Fear Experiences in Naturalistic Contexts: an Ecologically Valid fMRI Signature Integrating Brain Activation and Connectivity. *IEEE Trans. Affect. Comput.* 1–18 (2025) doi:10.1109/TAFFC.2025.3624391.
 25. Kringelbach, M. L., Perl, Y. S., Tagliazucchi, E. & Deco, G. Toward naturalistic neuroscience: Mechanisms underlying the flattening of brain hierarchy in movie-watching compared to rest and task. *Sci. Adv.* **9**, eade6049 (2023).
 26. Schultz, J. & Pilz, K. S. Natural facial motion enhances cortical responses to faces. *Exp. Brain Res.* **194**, 465–475 (2009).
 27. Zhang, R. *et al.* A neurofunctional signature of affective arousal generalizes across

- valence domains and distinguishes subjective experience from autonomic reactivity. *Nat. Commun.* **16**, 6492 (2025).
28. Morawetz, C., Hemetsberger, F. J., Laird, A. R. & Kohn, N. Emotion regulation: From neural circuits to a transdiagnostic perspective. *Neurosci. Biobehav. Rev.* **168**, 105960 (2025).
 29. Fernandez, K. C., Jazaieri, H. & Gross, J. J. Emotion Regulation: A Transdiagnostic Perspective on a New RDoC Domain. *Cogn. Ther. Res.* **40**, 426–440 (2016).
 30. Sloan, E. *et al.* Emotion regulation as a transdiagnostic treatment construct across anxiety, depression, substance, eating and borderline personality disorders: A systematic review. *Clin. Psychol. Rev.* **57**, 141–163 (2017).
 31. Stellern, J. *et al.* Emotion regulation in substance use disorders: a systematic review and meta-analysis. *Addiction* **118**, 30–47 (2023).
 32. Zilverstand, A., Parvaz, M. A. & Goldstein, R. Z. Neuroimaging cognitive reappraisal in clinical populations to define neural targets for enhancing emotion regulation. A systematic review. *Neuroimage* **151**, 105–116 (2017).
 33. Sun, J. *et al.* Practical AI application in psychiatry: historical review and future directions. *Mol. Psychiatry* **30**, 4399–4408 (2025).
 34. Koban, L., Wager, T. D. & Kober, H. A neuromarker for drug and food craving distinguishes drug users from non-users. *Nat. Neurosci.* **26**, 316–325 (2023).
 35. Xu, S. *et al.* Ecologically-Valid Emotion Signatures Enhance Mood Disorder Diagnostics. *Adv. Sci.* e05524 (2026) doi:10.1002/advs.202505524.
 36. Kronberg, G. & Goldstein, R. Z. An fMRI marker of drug and food craving. *Nat. Neurosci.* **26**, 178–180 (2023).
 37. Gross, J. J. Emotion Regulation: Current Status and Future Prospects. *Psychol. Inq.* **26**, 1–26 (2015).
 38. Gross, J. J. & Feldman Barrett, L. Emotion Generation and Emotion Regulation: One or Two Depends on Your Point of View. *Emot. Rev.* **3**, 8–16 (2011).